# Universal control of a six-qubit quantum processor in silicon

Stephan G. J. Philips[1,3], Mateusz T. Mądzik[1,3], Sergey V. Amitonov[1], Sander L. de Snoo[1], Maximilian Russ[1], Nima Kalhor[1], Christian Volk[1], William I. L. Lawrie[1], Delphine Brousse[2], Larysa Tryputen[2], Brian Paquelet Wuetz[1], Amir Sammak[2], Menno Veldhorst[1], Giordano Scappucci[1] & Lieven M. K. Vandersypen[1✉]

Future quantum computers capable of solving relevant problems will require a large number of qubits that can be operated reliably[1]. However, the requirements of having a large qubit count and operating with high fidelity are typically conflicting. Spins in semiconductor quantum dots show long-term promise[2,3] but demonstrations so far use between one and four qubits and typically optimize the fidelity of either single- or two-qubit operations, or initialization and readout[4–11]. Here, we increase the number of qubits and simultaneously achieve respectable fidelities for universal operation, state preparation and measurement. We design, fabricate and operate a six-qubit processor with a focus on careful Hamiltonian engineering, on a high level of abstraction to program the quantum circuits, and on efficient background calibration, all of which are essential to achieve high fidelities on this extended system. State preparation combines initialization by measurement and real-time feedback with quantum-non-demolition measurements. These advances will enable testing of increasingly meaningful quantum protocols and constitute a major stepping stone towards large-scale quantum computers.

On the path to practical large-scale quantum computation, electron spin qubits in semiconductor quantum dots[12] show promise because of their inherent potential for scaling through their small size[13,14], long-lived coherence[4] and compatibility with advanced semiconductor manufacturing techniques[15]. Nevertheless, spin qubits currently lag behind in scale when compared to superconducting, trapped ions and photonic platforms, which have demonstrated control of several dozen qubits[16–18]. By comparison, using semiconductor spin qubits, partial[19] and universal[11] control of up to four qubits was achieved and entanglement of up to three qubits was quantified[9,10,20]. In a six-dot linear array, two qubits encoded in the state of three spins each were operated[21] and spin exchange oscillations in a 3 × 3 array have been reported[22].

Furthermore, the experience with other qubit platforms shows that, in scaling up, maintaining the quality of the control requires substantial effort, particularly, for instance, to deal with the denser motional spectrum in trapped ions[23], to avert crosstalk in superconducting circuits[24] or to avoid increased losses in photonic circuits[25]. For small semiconductor spin qubit systems, state-of-the-art single-qubit gate fidelities exceed 99.9%[5,26,27] and two-qubit gates well above 99% fidelity have been demonstrated recently[6–8,10]. Most quantum-dot-based demonstrations suffer from low initialization or readout fidelities, with typical visibilities of no more than 60–75%, with only a few recent exceptions[8,21,28]. Conversely, high-fidelity spin readout has been claimed on the basis of an analysis of the readout error mechanisms, but these claims have not been validated in combination with high-fidelity qubit control[29,30]. Although high-fidelity initialization, readout, single-qubit gates and two-qubit gates have thus been demonstrated individually in

small systems, almost invariably one or more of these parameters are appreciably compromised while optimizing others. A major challenge and important direction for the field is therefore to achieve high fidelities for all components while at the same time enlarging the qubit count.

Here we study a system of six spin qubits in a linear quantum dot array and test what performance can be achieved using known methods, such as multi-layer gate patterns for independent control of the two-qubit exchange interaction[31–33] and micromagnet gradients for electric-dipole spin resonance (EDSR) and selective qubit addressing[34]. Furthermore, we introduce several new techniques for semiconductor qubits that, collectively, are critical for the improvement of the results and help facilitate scalability, such as initialization by measurement using real-time feedback[35], qubit initialization and measurement without reservoir access, and efficient calibration routines. Initialization and readout circuits span the full six-qubit array. We characterize the quality of the control by preparing maximally entangled states of two and three spins across the array.

The six-qubit array is defined electrostatically in the [28]Si quantum well of a [28]Si/SiGe heterostructure, between two sensing quantum dots, as seen in Fig. 1a (Methods). The multi-layer gate pattern enables excellent control of the charge occupation of each quantum dot and of the tunnel couplings between neighbouring quantum dots. These parameters are controlled independently through linear combinations of gate voltages, known as virtual gates[36]. The interdot pitch is chosen to be 90 nm, which for this 30-nm-deep quantum well yields easy access to the regime with one electron in each dot, indicated for short as the (1,1,1,1,1,1) charge occupation. Low valley splittings on

[1]QuTech and the Kavli Institute of Nanoscience, Delft University of Technology, Delft, the Netherlands. [2]QuTech and Netherlands Organization for Applied Scientific Research (TNO), Delft, the Netherlands. [3]These authors contributed equally: Stephan G. J. Philips, Mateusz T. Mądzik. ✉e-mail: L.M.K.Vandersypen@tudelft.nl

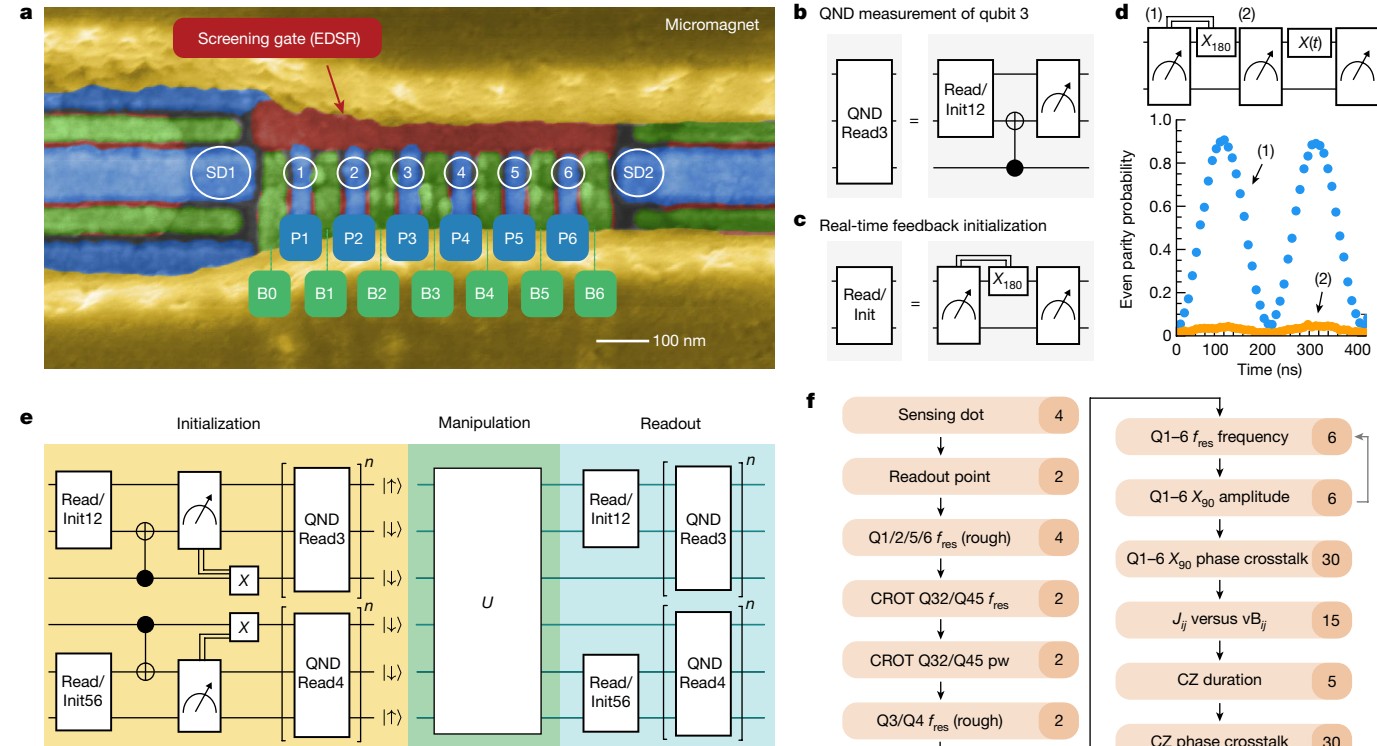

**Fig. 1 | Device initialization, measurement and calibration. a**, A false-coloured scanning electron microscope image of a device similar to the one used in the experiments. Each colour represents a different metallization layer. Plunger (P, blue) and barrier (B, green) gates are used to define quantum dots in the channel between the screening gates (red) and sensing dots (SD1 and SD2) on the side. Two cobalt micromagnets (yellow) are placed on top of the gate stack. **b,c**, Buildings blocks used for readout (read) and initialization (init) in this experiment, showing the circuit used to perform a single QND measurement of qubit Q3 (**b**) and the circuit used for spin measurement and initialization using a parity measurement (**c**). The double line in the diagram indicates that $X_{180}$ rotation is conditional on the measurement outcome. **d**, An example of a CROT used to initialize the qubits. The sequence shown is applied repeatedly with short time intervals, with the final state of one cycle being the initial state of the next. (1) shows the even parity probability of the first measurement; (2) shows the even parity probability after the bit flip conditional on the first measurement outcome. **e**, Schematic showing the total scheme used for the initialization and readout of all six qubits, with $U$, the unitary matrix of the manipulation stage (see Extended Data Fig. 2 for an expanded view). **f**, Calibration graph used in the experiments. The numbers on the right show the number of parameters that are calibrated in each step.

Si/SiGe devices have hindered progress in the past[37], but in this device all valley splittings are in the range of 100–300 μeV (Supplementary Information).

In designing the qubit measurement scheme, we focused on achieving short measurement cycles in combination with high-fidelity readout, as this accelerates testing of all other aspects of the experiment. To measure the outer qubit pairs, we use Pauli spin blockade (PSB) to probe the parity of the two spins (rather than differentiating between singlet and triplet states), exploiting the fact that the $T_0$ triplet relaxes to the singlet well before the end of the 10 μs readout window. We tune the outer dot pairs of the array to the (3,1) electron occupation, where the readout window is larger than in the (1,1) regime (Extended Data Fig. 1). As the sensing dots are less sensitive to the charge transition between the centre dots, the middle qubits are measured by quantum-non-demolition (QND) measurements that map the state of qubit Q3(Q4) on qubit Q2(Q5) through a conditional rotation (CROT) (Fig. 1b)[5,38]. In this way, for every iteration of the experiment, 4 bits of information are retrieved that depend on the state of all six physical qubits. Iterative operation permits full readout of the six-qubit system.

Qubit initialization is based on measurements of the spin state across the array followed by real-time feedback to place all qubits in the target initial state. This scheme has the benefit of not relying on slow thermalization and that no access to electron reservoirs is needed to bring in fresh electrons, which is helpful for scaling to larger arrays. In fact, we had experimental runs of more than one month in which the electrons stayed within the array continuously. For qubits Q3 and Q4, real-time feedback simply consists of flipping the qubit if the measurement returned |↑⟩. Initialization of qubits Q1 and Q2 (or Q5 and Q6) using parity measurements and real-time feedback is illustrated in Fig. 1c. First, assuming that the qubits start from a random state, we perform a parity measurement that will cause the state to either collapse to an even (|↓↓⟩, |↑↑⟩) or odd (|↑↓⟩/|↓↑⟩) parity (Methods). After the measurement, a π pulse is applied to qubit Q1 in case of even parity, which converts the state to odd parity (feedback latency 660 ns). Subsequently, we perform a second measurement, which converts either of the odd parity states to |↑↓⟩. Specifically, when pulsing towards the readout operating point, both |↑↓⟩ and |↓↑⟩ relax into the singlet state ((4,0) charge occupation). When pulsing adiabatically from the (4,0) back to the (3,1) charge configuration, the singlet is mapped onto the |↑↓⟩ state. If the qubit initialization is successful, the second measurement should return an odd parity (with typically around 95% success rate). To further boost the initialization fidelity we use the outcome of the second measurement to postselect successful experiment runs (Extended Data Fig. 1d). Figure 1d shows initialization by measurement of the first two qubits. The first readout outcome (blue) shows Rabi oscillations controlled by a microwave burst of variable duration applied near the end of the previous cycle (see Methods for more details). The second readout outcome (green) shows the state after the real-time classical feedback step. The oscillation has largely vanished, indicating successful initialization by measurement and feedback.

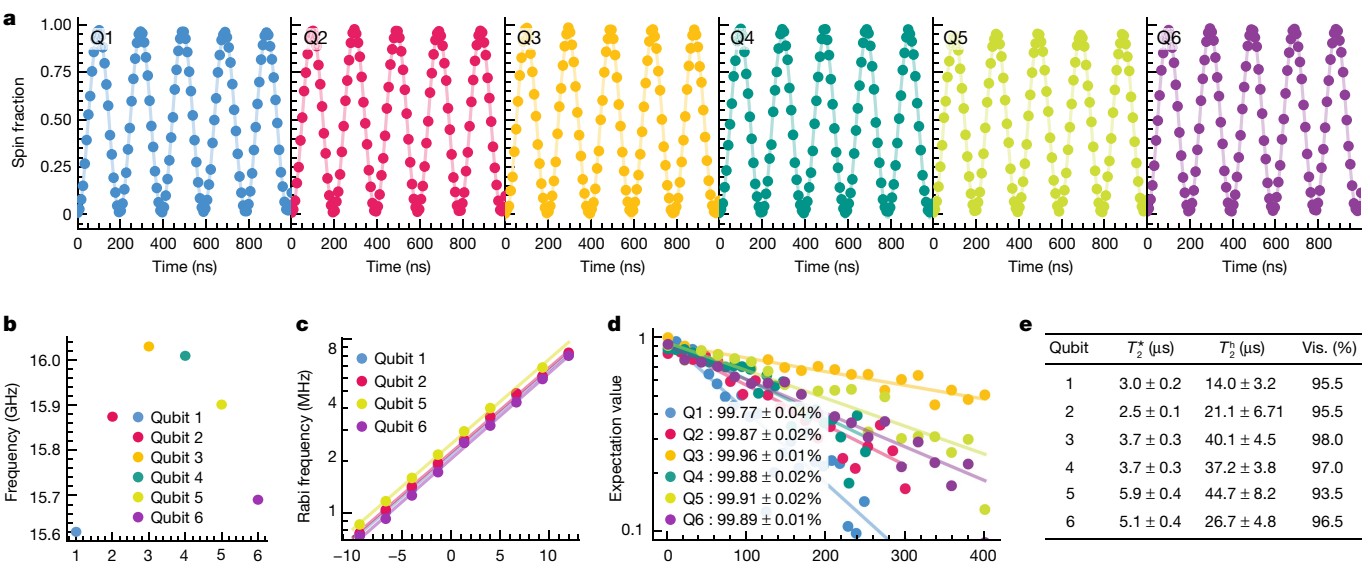

**Fig. 2 | Single-qubit gate characterization. a**, Rabi oscillations for every qubit, taken sequentially. The spin fraction refers to the spin-up fraction for qubits Q2–Q5 and to the spin-down fraction for qubits Q1 and Q6. The drive amplitudes were adjusted in order to obtain uniform Rabi frequencies of 5 MHz. **b**, Qubit frequency for each of the six qubits. **c**, Rabi frequency of each qubit as a function of the applied microwave power. **d**, Randomized benchmarking results for each qubit, using a 5 MHz Rabi frequency. The reported fidelity is the average single-qubit gate fidelity, and the uncertainties ($2\sigma$) are calculated using the covariance matrix of the fit. **e**, Table showing the dephasing time $T_2^*$, Hahn echo decay time ($T_2^\mathrm{h}$) and visibilities (vis.) for each qubit.

The sequence to initialize and measure all qubits is shown in Fig. 1e (see Extended Data Fig. 2 for the unfolded quantum circuit). We sequentially initialize qubit pair Q5 and Q6, then qubit Q4, then qubits Q1 and Q2, and finally qubit Q3, using the steps described above (for compactness, the steps appear as being simultaneous in the diagram). In order to further enhance the measurement and initialization fidelities, we repeat the QND measurement three times, alternating the order of the qubit Q3 and Q4 measurements. We postselect runs with three identical QND readout outcomes in both the initialization and measurement steps (except for Fig. 5 below, where readout simply uses majority voting). After performing the full initialization procedure depicted in Fig. 1e, the six-qubit array is initialized in the state $|\uparrow\downarrow\downarrow\downarrow\downarrow\uparrow\rangle$. In all measurements below, we initialize either two, three or all six qubits, depending on the requirement of the specific quantum circuit we intend to run. We leave the unused qubits randomly initialized, as the visibilities decrease when initializing all six qubits within a single shot sequence (Extended Data Fig. 3). When operating on individual qubits, the initialization and measurement procedures yield visibilities of 93.5–98.0% (Fig. 2e). To put these numbers in perspective, if the readout error for both $|0\rangle$ and $|1\rangle$ were 1% alongside an initialization error of 1%, the visibility would be 96%.

We manipulate the qubits by EDSR[39]. A micromagnet located above the gate stack is designed to provide both qubit addressability and a driving field gradient (Fig. 1a and Supplementary Information). We can address each qubit individually and drive coherent Rabi oscillations as depicted in Fig. 2. We observe no visible damping in the first five periods. The data in Fig. 2b shows that the qubit frequencies are not spaced linearly, deviating from our prediction based on numerical simulations of the magnetic field gradients (Supplementary Fig. 1). However, the smallest qubit frequency separation of approximately 20 MHz is sufficient for selective qubit addressing with our operating speeds varying between 2 and 5 MHz. The Rabi frequency is linear in the driving amplitude over the typical range of microwave power used in the experiment (Fig. 2c). We operate single-qubit gates sequentially, to ensure we stay in this linear regime and to keep the calibration simple. Simultaneous rotations would involve additional

characterization and compensation of crosstalk effects (see also Extended Data Fig. 5). We characterize the single-qubit properties of each qubit separately. Figure 2d shows the results of randomized benchmarking experiments. All average single-qubit gate fidelities are between $99.77 \pm 0.04\%$ and $99.96 \pm 0.01\%$, which demonstrates that, even within this extended qubit array, we retain high-fidelity single-qubit control. The coherence times of each qubit are tabulated in Fig. 2e. We expect spin coherence to be limited by charge noise coupled in by the micromagnet[40].

Two-qubit gates are implemented by pulsing the (virtual) barrier gate between adjacent dots while staying at the symmetry point. Pulsing the barrier gate leads to a ZZ interaction (throughout, *X*, *Y* and *Z* stand for the Pauli operators, *I* for the identity and *ZZ* is shorthand for the tensor product of two Pauli *Z* operators, and so on), given that the effect of the flip-flop terms of the spin exchange interaction is suppressed because of the differences in the qubit splittings[41]. The quantum circuit in Fig. 3a measures the time evolution under the *ZZ* component of the Hamiltonian only, as the single-qubit $\pi$ pulses in between the two exchange pulses decouple any *IZ/ZI* terms[42]. The measured signal oscillates at a frequency $J/2$ (Fig. 3b–f) as a function of the barrier gate pulse duration, corresponding to controlled phase (CPhase) evolution. When pulsing only the barrier gate between the target qubit pair, the desired on/off ratio of $J_{ij}$ (>100) could not be achieved. We solve this, without sacrificing operation at the symmetry point, by using a linear combination of the virtual barrier gates (vB1–vB6). Specifically, the barrier gates around the targeted quantum dot pair are pulsed negatively to push the corresponding electrons closer together and thereby enhance the exchange interaction (Extended Data Table 1). The exponential dependence of $J_{ij}$ on the virtual barrier gates is seen in Fig. 3h. In Fig. 3g we investigate the residual exchange of idle qubit pairs, while one qubit pair is pulsed to its maximal exchange value within the operating range. The results show minimal residual exchange amplitudes in the off state between the other pairs.

Through suitable timing, we use the CPhase evolution to implement a controlled-*Z* (CZ) gate. Figure 3j shows the pulse shape that is used to ensure a high degree of adiabaticity throughout the CZ gate[6]. We

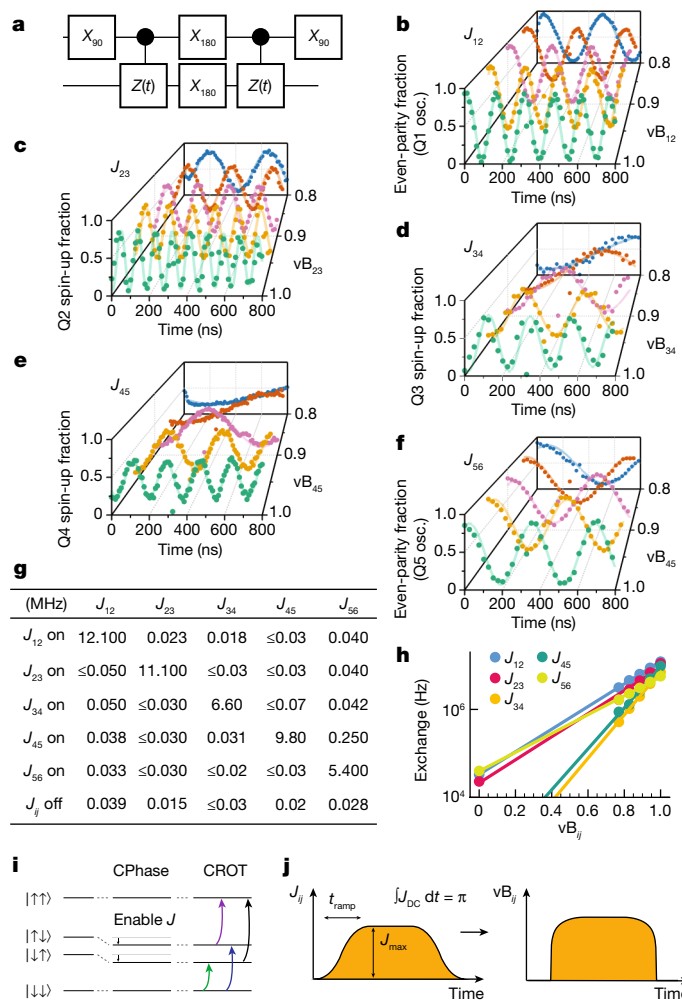

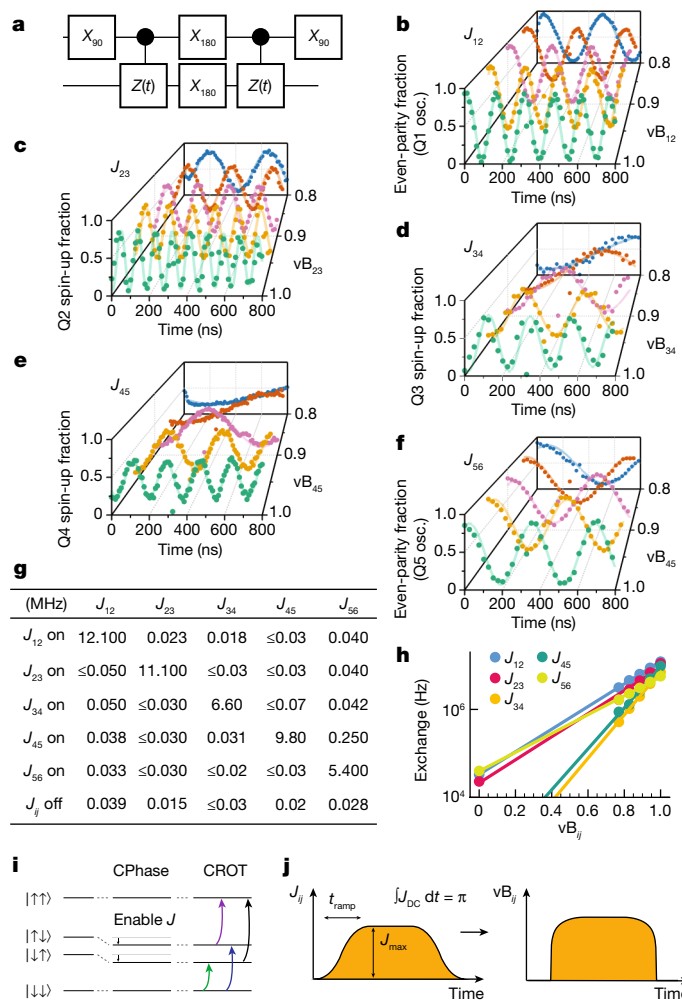

**Fig. 3 | Two-qubit gate characterization. a**, Quantum circuit used to measure CPhase oscillations between a pair of qubits. **b–f**, Measured spin probabilities as a function of the total evolution time 2$t$ for neighbouring qubit pairs Q1–Q2 (**b**), Q2–Q3 (**c**), Q3–Q4 (**d**), Q4–Q5 (**e**) and Q5–Q6 (**f**) for different virtual barrier gate voltages (with 0 and 1 corresponding to the exchange switched off and at its maximum value). **g**, Maximum exchange coupling measured for each qubit pair, and the corresponding residual exchange coupling for the other pairs, achievable within the AWG pulsing range without retuning of the static gate voltages. Bottom row: $J_{ij}$ with all exchange couplings switched off (see Supplementary Information for error bars). **h**, Exchange coupling versus virtual barrier gate voltage for all qubit pairs. **i**, Schematic showing the energy levels in the absence (left) and presence (middle, right) of the effective Ising $ZZ$ interaction under exchange (see text). Owing to the $ZZ$ coupling term, the antiparallel spin states are lowered in energy, and pick up an additional phase as a function of time, resulting in a CPhase evolution. The shifted energy levels also enable conditional microwave-driven rotations (CROT), which we use during initialization and readout. **j**, Pulse shape of the exchange amplitude throughout a gate voltage pulse used for the CZ gate, and the corresponding pulse shape converted to gate voltage.

use a Tukey window as waveform, with a ramp time of $\tau_{\text{ramp}} = \frac{3}{\sqrt{\delta B^2 + J_{\text{max}}^2}}$ (ref. [43]). This pulse shape is defined in units of energy and we convert it into barrier voltages using the measured voltage to exchange the energy relation[6].

One of the challenges when operating larger quantum processors is to track and compensate for any dynamical changes in qubit parameters to ensure high-fidelity operation, initialization and readout. Another challenge is to keep track of and compensate for crosstalk effects imparted by both single- and two-qubit gates on the phase evolution of each qubit. We perform automated calibrations, as shown in

Fig. 1f, and correct 108 parameters in total. The detailed description of each calibration routine is included in the Methods and Extended Data Fig. 4. Twice a week, we run the full calibration scheme, which takes about one hour. Every morning, we run the calibration scheme leaving out the phase corrections for single-qubit operations and the dependence of $J_{ij}$ on the virtual barrier gates $vB_{ij}$. Sometimes, specific calibrations, especially qubit frequencies and readout coordinates, are re-run throughout the day, as needed. Supplementary Fig. 3 plots the evolution of the calibrated values for a number of qubit parameters over the course of one month.

With single- and two-qubit control established across the six-qubit array, we proceed to create and quantify pairwise entanglement across the quantum dot array as a measure of the quality of the qubit control (Fig. 4a–e). These experiments benefited from a high level of abstraction in the measurement software, allowing us to flexibly program a variety of quantum circuits acting on any of the qubits, drawing on the table of 108 calibration parameters that is kept updated in the background and on the detailed waveforms to achieve high-fidelity gates. The parity readout of the outer qubits yields a native $ZZ$ measurement operator. We measure single-qubit expectation values by mapping the $ZZ$ operator to a $ZI/IZ$ operator, as shown in Fig. 4g. This allows full reconstruction of the density matrix. The state fidelity is calculated using $F = \langle \psi | \rho | \psi \rangle$, where $\psi$ is the target state and $\rho$ is the measured density matrix. The target states are maximally entangled Bell states. The obtained density matrices measured across the six-dot array have state fidelity ranging from 88% to 96%, which is considerably higher than the Bell state fidelities of 78% to 89% (all state preparation and measurement (SPAM) corrected, see Methods) reported on two-qubit quantum dot devices just a few years ago[42,44,45].

As a final characterization of the qubit control across the array, we prepare Greenberger–Horne–Zeilinger (GHZ) states, which are the most delicate entangled states of three qubits[46,47]. Figure 5a shows the quantum circuit we used to prepare the GHZ states. The full circuit, including initialization and measurement, contains up to 14 CROT operations, 2 CZ operations, 42 parity measurements, 16 single-qubit rotations conditional on real-time feedback and 5 single-qubit $X_{90}$ rotations (Extended Data Fig. 2). The measurement operators for quantum state tomography are generated in a similar manner as for the Bell states. In order to reconstruct three-qubit density matrices, we perform measurements in 26 (for qubits Q2–Q3–Q4 and Q3–Q4–Q5) or 44 (for qubits Q1–Q2–Q3 and Q4–Q5–Q6) different basis and repeat each set 2,000 times to collect statistics. A full dataset consisting of 52,000 (88,000) single-shot repetitions takes about 5 min to acquire, thanks to the efficient uploading of waveforms to the waveform generator (Methods) and the short single-shot cycle times. Figure 5b–e shows the measured density matrices for qubits Q1–Q2–Q3, Q2–Q3–Q4, Q3–Q4–Q5 and Q4–Q5–Q6. The obtained state fidelities range from 71% to 84% (see Methods for a brief discussion of dephasing effects from heating). For comparison, the record GHZ state fidelity reported recently for a triple quantum dot spin qubit system is 88% (ref. [9]). The same dataset from ref. [9] analysed without readout correction yields 45.8% fidelity, whereas our results with no readout error removal range from 52.8% to 67.2% (Supplementary Information). The reduction in state fidelities compared to the two-qubit case (especially when involving qubits Q3 and Q4) is mainly due to increased SPAM errors. From the same data sets, we calculate entanglement witnesses, which clearly demonstrate three-qubit entanglement (Supplementary Information).

The demonstration of universal control of six qubits in a $^{28}$Si/SiGe quantum dot array advances the field in multiple ways. While scaling to a record number of qubits for a quantum dot system, we achieve Rabi oscillations for each qubit with visibilities of 93.5–98.0%, implying high readout and initialization fidelities. The initialization uses a new scheme relying on qubit measurement and real-time feedback. Readout relies on PSB and QND measurements. This combination of initialization and readout allows the device to be operated while

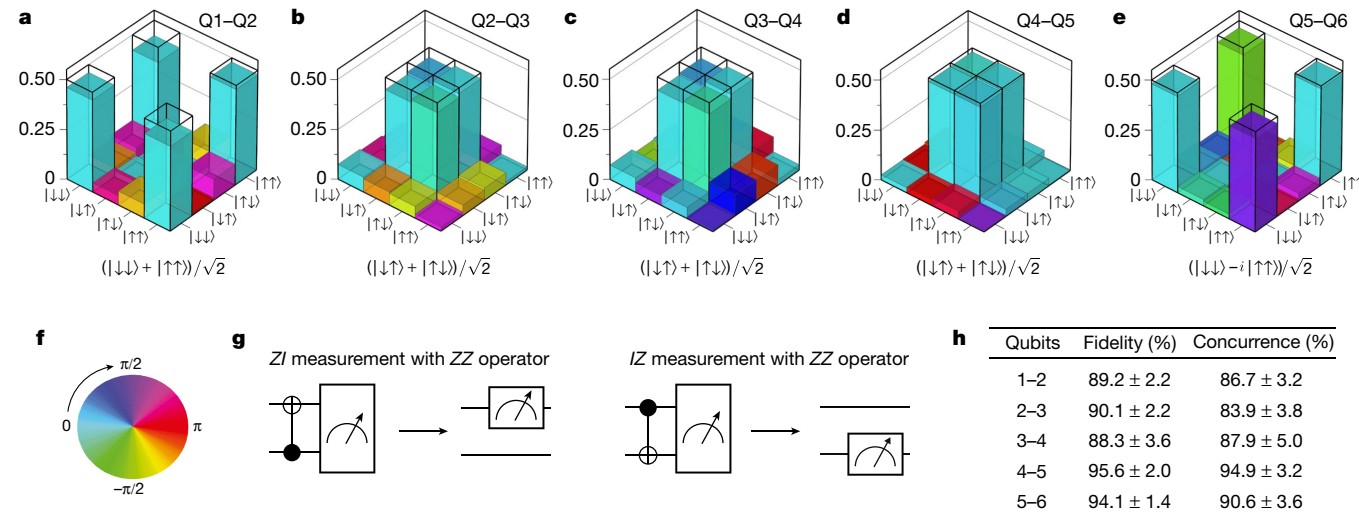

**Fig. 4 | Bell state tomography. a–e**, Measured two-qubit density matrices for qubits Q1–Q2 (**a**), Q2–Q3 (**b**), Q3–Q4 (**c**), Q4–Q5 (**d**) and Q5–Q6 (**e**), after removal of SPAM errors (see Supplementary Information for the uncorrected density matrices). The target Bell states are indicated and outlined with the wireframes. **f**, Colour wheel with phase information for the density matrices presented in **a–e**. **g**, Quantum circuits used for converting parity readout (*ZZ*) into effective single-qubit readout (*IZ* and *ZI*). **h**, State fidelities of the measured density matrices with respect to the target Bell states and the concurrences for the measured density matrices. Error bars (2σ) are derived from Monte Carlo bootstrap resampling[9,44,59]. State fidelities without readout error removal: qubits Q1–Q2, 88.2%; Q2–Q3, 83.8%; Q3–Q4, 78.0%; Q4–Q5, 91.3%; Q5–Q6, 91.3%.

retaining the six electrons in the linear quantum dot array, alleviating the need for access to electron reservoirs. All single-qubit gate fidelities are around 99.9% and the high quality of the two-qubit gates can be inferred from the 89–95% fidelity Bell states prepared across the array. The development of a modular software stack, efficient calibration routines and reliable device fabrication have been essential for this experiment. Future work must focus on understanding and mitigating heating effects leading to frequency shifts and reduced dephasing times, as we find this to be the limiting factor in executing complicated quantum circuits on many qubits. The use of simultaneous single-qubit rotations and simultaneous two-qubit CZ gates will keep pulse sequences more compact, at the expense of additional

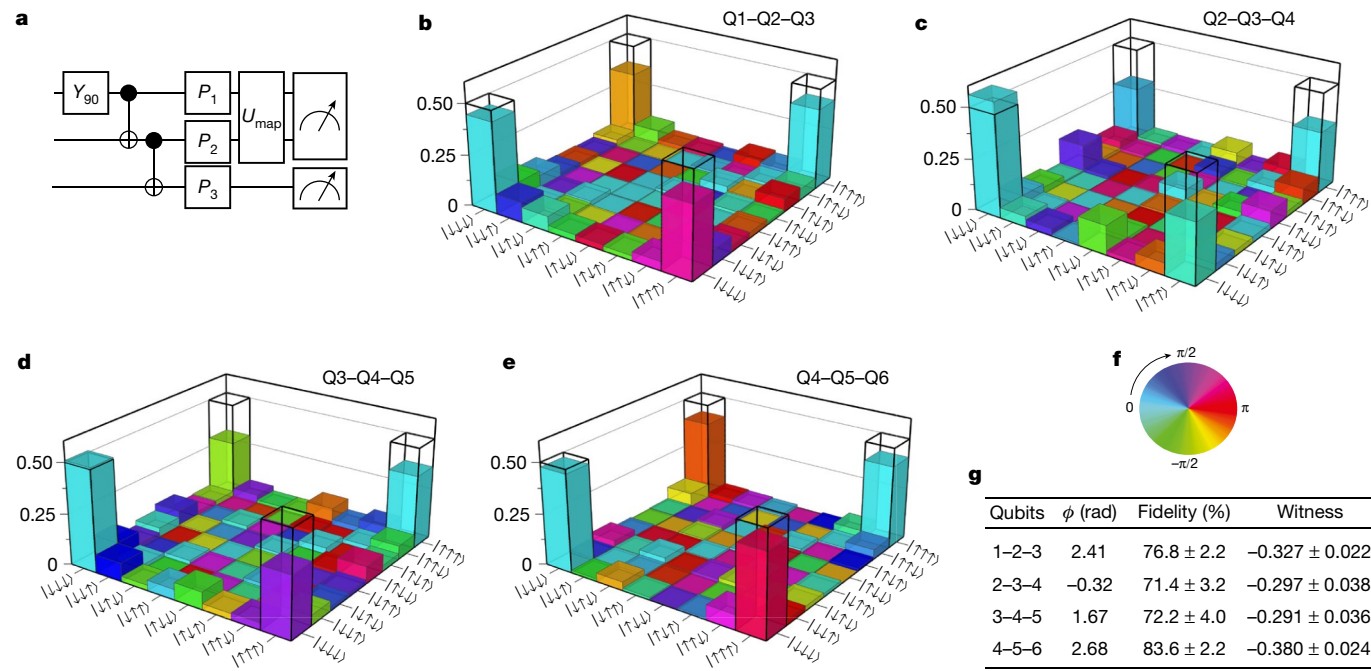

**Fig. 5 | Three-qubit GHZ state tomography. a**, Circuit diagram used to prepare the GHZ states. The $U_{map}$ operation is the unitary that is executed in case we measure the *IZ* or *ZI* projections on qubits Q1–Q2 and Q5–Q6, similar to the Bell state experiments. **b–e**, Density matrices of the prepared GHZ states using qubits Q1–Q2–Q3 (**b**), Q2–Q3–Q4 (**c**), Q3–Q4–Q5 (**d**) and Q4–Q5–Q6 (**e**), obtained using quantum state tomography, after removal of SPAM errors (see Supplementary Information for the uncorrected density matrices). The black wireframes correspond to the ideal GHZ state. **f**, Colour wheel with phase information for the density matrices presented in **d** and **e**. **g**, Table showing the state fidelities and entanglement witness values for the different qubit sets. We choose $\phi$ in $|\psi_{GHZ}\rangle = (|000\rangle + e^{i\phi}|111\rangle)\sqrt{2}$, with respect to the highest state fidelity. State fidelities without SPAM removal: qubits Q1–Q2–Q3, 64.3%; Q2–Q3–Q4, 52.8%; Q3–Q4–Q5, 52.7%; Q4–Q5–Q6, 67.2%.

calibrations. This will require accounting for crosstalk effects, which we anticipate will be easiest for the two-qubit gates. We estimate that the concepts used here for control, initialization and readout can be used without substantial modification in arrays that are twice as long, as well as in small two-dimensional arrays (Supplementary Information). Scaling further will require additional elements such as cross-bar addressing to control dense two-dimensional arrays[48,49] and on-chip quantum links to connect local quantum registers together[3,50–52].

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

# Methods

## Device fabrication

Devices are fabricated on an undoped $^{28}$Si/SiGe heterostructure featuring an 8 nm strained $^{28}$Si quantum well, with a residual $^{29}$Si concentration of 0.08%, grown on a strain-relaxed $Si_{0.7}Ge_{0.3}$ buffer layer. The quantum well is separated from the surface by a 30-nm-thick $Si_{0.7}Ge_{0.3}$ spacer and a sacrificial 1 nm Si capping layer. The gate stack consists of three layers of Ti:Pd metallic gates (3:17, 3:27 and 3:27 nm) isolated from each other by 5 nm $Al_2O_3$ dielectrics, deposited using atomic layer deposition. A ferromagnetic Ti:Co (5:200 nm) layer on top of the gate stack creates a local magnetic field gradient for qubit addressing and manipulation. The ferromagnetic layer is isolated from the gate layers by 10 nm of $Al_2O_3$ dielectric. The cobalt layer is not covered with a dielectric. Further details of device fabrication methods can be found in ref. [33].

After fabrication, all the devices are screened at 4 K. We check for current leakage, accumulation below the gates and device stability (for example, drifts in current). The best device (if it meets our requirements) is selected and cooled down in a dilution refrigerator. The fraction of devices that pass these 4 K checks varies from 0 to 50% per batch (a batch contains either 12 or 24 devices).

## Microwave crosstalk and synchronization condition

In Fig. 2, the single-qubit gates are chosen to be operated at a 5 MHz Rabi frequency and all single-qubit randomized benchmarking results are taken at this frequency as well. When operating all qubits within the same sequence, we were unable to operate at a 5 MHz Rabi frequency as qubits Q2(Q3) and Q5(Q4) are too close to each other in frequency. We used the synchronization condition[53,54] to choose Rabi frequencies for the single-qubit gates for which the qubit that suffers crosstalk does not undergo a net rotation while the target qubit is rotated by 90 degrees or multiples thereof (Extended Data Fig. 5). The Rabi frequencies for the state tomography experiments are as follows (qubits Q1–Q6): 4.6 MHz, 1.9 MHz, 4.2 MHz, 3.6 MHz, 2.4 MHz and 5 MHz.

## Automated calibration routines

Calibrations are a crucial part in operating a multi-qubit device. Figure 1f lists the necessary calibration types that need to be corrected periodically and Extended Data Fig. 4 shows an example calibration for each parameter type. Every calibration uses an automated script to extract the optimal value for the measured parameter, which is recorded in a database. In our framework, the operator chooses to accept this value or to re-run the calibration.

**Sensing dot (5 s).** The calibrations routine starts by calibrating the sensing dots (Extended Data Fig. 4a) to the most sensitive operating point for parity mode PSB readout. We scan the (virtual) plunger voltage of the sensing dot for two different charge configurations of the corresponding double dot, corresponding to the singlet and triplet states. One configuration is in the (3,1) region and the other in the (4,0) region, in order to be insensitive to small drifts in the gate voltages. The calibration returns the plunger voltage for which the largest difference is obtained in the sensing dot signal between these two cases (Extended Data Fig. 4a). From this difference, we also set the threshold in the demodulated in-phase (I) and quadrature (Q) signals (in short, IQ signals) of the radio frequency (RF) modulated readout, to allow singlet/triplet differentiation (the IQ signal is converted to a scalar by adjusting the phase of the signal). The threshold is chosen halfway between the signals for the two charge configuration. During qubit manipulation, the sensing dot is kept in Coulomb blockade. It is only pulsed to the readout configuration when executing the readout.

**Readout point (35 s).** The parity mode PSB readout is calibrated by finding the optimal voltage of the plunger gates near the anticrossing for the readout. The readout point is only calibrated along one axis (vP1

or vP5), for simplicity, and as the performance of the PSB readout is similar at any location along the anticrossing. In the calibration shown in Extended Data Fig. 4b, we initialize either a singlet ($|\uparrow\downarrow\rangle$) or a triplet ($|\downarrow\downarrow\rangle$, using a single-qubit gate) state and sweep the plunger gate to find to the optimal readout point.

**Resonance frequency of qubits Q1, Q2, Q5 and Q6 (rough) (17 s).** We perform a course scan of the resonance frequencies of qubits Q1, Q2, Q5 and Q6 (Extended Data Fig. 4c) around the previously saved values. We fit the Rabi formula

$$P_s(t) = \frac{\Omega^2}{\Omega^2 + \Delta^2}\sin^2\left(\frac{\sqrt{\Omega^2 + \Delta^2}}{2}t\right) \tag{1}$$

to the experimental data and extract the resonance frequency, where $P_s(t)$ is the spin probability, $\Omega$ is the Rabi frequency, and $\Delta$ is the frequency difference between the resonance frequency of the qubit and the applied microwave tone.

**QND readout: CROT Q32, Q45 resonance frequency (14 s).** Subsequently, we calibrate the QND readout for qubits Q3 and Q4. To perform QND readout, we need to calibrate a CROT gate. We choose to use a controlled rotation two-qubit gate, as it requires little calibration (compared to the CPhase) given that we can ignore phase errors during readout.

We set the exchange to 10–20 MHz by barrier gate pulses and scan the CROT driving frequency (Extended Data Fig. 4d) around the previously saved values. Again, we fit the Rabi formula in equation (1) to extract the optimal resonance frequency.

**QND readout: CROT Q32, Q45 pulse width (25 s).** Next, we tune the optimal microwave burst duration for the CROT gate, by driving Rabi oscillations (Extended Data Fig. 4e) in the presence of the exchange coupling. We fit the decaying sinusoid

$$P_s(t) = \frac{A}{2}\sin(\omega t - \phi_0)e^{-\frac{t}{\tau}} + B \tag{2}$$

and extract the pulse width the for CROT gate.

**Resonance frequency of qubits Q3 and Q4 (rough) (28 s).** With QND readout established, we scan the driving frequency for qubits Q3 and Q4 in a similar manner as we did for Q1, Q2, Q5, Q6 (Extended Data Fig. 4f). The calibration scripts will automatically use QND readout for Q3 and Q4 calibration, in place of the PSB readout for Q1, Q2, Q5 and Q6.

**Resonance frequency and amplitude (fine) (Q1, Q2, Q5, Q6→frequency 22 s, amplitude 23 s; Q3, Q4→frequency 32 s, amplitude 34 s).** We calibrate more accurately the qubit frequency and driving amplitude using an error amplification sequence (Extended Data Fig. 4g,h), where we execute an $X_{90}$ gate 18 times and sweep either the frequency or the amplitude of the microwave burst. We fit the data using the Rabi formula in equation (1) once again to extract the resonancy frequency. The amplitude of the microwave burst is controlled by the IQ input channels of the vector source we used. To calibrate the amplitude for an $X_{90}$ rotation, we vary the amplitude applied to the IQ input and fit the result to a Gaussian function,

$$P_s(x) = \alpha e^{-\frac{(x-\mu)^2}{2\sigma^2}} . \tag{3}$$

where $x$ is the input amplitude of the IQ signal, $\mu$ is the centre of the peak (optimal amplitude) and $\sigma$ is the peak width. This functional form is not strictly correct but it does find the optimal amplitude for an $X_{90}$ rotation. We suspect that the longer amplification sequences gave

better results, as they more closely resemble the sequence lengths used for randomized benchmarking (including some 'heating effects').

In these calibrations, we only calibrate the $X_{90}$ gate. The $Y_{90}$ gate is implemented similarly to the $X_{90}$, but phase shifted. $Z$ gates are performed in software by shifting the reference frame. $X_{180}$ and $Y_{180}$ rotations are performed by applying two 90 degree rotations. We do not simultaneously drive two or more qubits.

**$X_{90}$ phase crosstalk (Q1, Q2, Q5, Q6→27 s; Q3, Q4→45 s).** Any single-qubit gate causes the Larmor frequency of the other qubits to shift slightly because of the applied microwave drive. We compensate for this by applying a virtual $Z$ rotation to every qubit after a single-qubit gate has been performed. The Ramsey-based sequence is used to calibrate the required phase corrections (Extended Data Fig. 4i) and data is fitted with the equation

$$P_s(\phi) = -\frac{A}{2}\cos(\phi - \phi_0) + B \quad (4)$$

where $A$ and $B$ are fitting parameters that correct for the limited visibility of the spin readout, $\phi$ is the applied virtual Z rotations in the calibration and $\phi_0$ is the fitted phase correction. A single $X_{90}$ pulse on one qubit will impart phase errors on qubits Q2 to Q6. Thus we need to calibrate separately 30 different phase factors, five for each qubit.

**$J_{ij}$ versus v$B_{ij}$ (qubit pairs Q12, Q56→146 s; qubit pairs Q23, Q45→207 s; qubit pair Q34→299 s).** Two-qubit gates are implemented by applying a voltage pulse that increases the tunnel coupling between the respective quantum dots. To enable two-qubit gates, we take the following elements into account:

- Exchange strength. We operate the two-qubit gates at exchange strengths $J_{on}$ where the quality factor of the oscillations is maximal. This condition is found for $J_{on} \approx 5$ MHz.
- Adiabacity condition. When the Zeeman energy difference ($\Delta E_z$) and the exchange ($J(t)$) are of the same order of magnitude, care has to be taken to maintain adiabaticity throughout the CPhase gate. We do this by applying a Tukey-based pulse, where the ramp time is chosen as $\tau_{ramp} = \frac{3}{\sqrt{\Delta E_z^2 + J_{on}^2}}$ (ref. [43]).
- Single-qubit phase shifts. As we apply the exchange pulse, the qubits will physically be slightly displaced. This causes a frequency shift and hence phase accumulation, which needs to be corrected for.

In order to satisfy these conditions, we need to know the relationship between the barrier voltage and the exchange strength. We construct this relation by measuring the exchange strength (Fig. 3a) for the last 25% of the virtual barrier pulsing range ($J > 1$ MHz regime). We fit the exchange to an exponential and extrapolate this to any exchange value (Extended Data Fig. 4j). This allows us to generate the adiabatic pulse as described in the main text and choose the target exchange value.

**CZ duration (qubit pairs Q12, Q56→29 s; qubit pairs Q23, Q45→34 s; qubit pair Q34→45 s).** The gate voltage pulse to implement a CZ operation uses a Tukey shape in $J$ by inverting the relationship $J(vB_{ij})$. The maximum value of $J$ is capped at $J_{on}$. The actual largest value of $J$ used and the length of the pulse then determine the phase acquired under $ZZ$ evolution. We first analytically evaluate the accumulated $ZZ$ evolution as a function of these parameters around the target of π evolution under $ZZ$, and then experimentally fine tune the actual accumulated $ZZ$ evolution by executing a Ramsey circuit with a decoupled CPhase evolution in between the two π/2 rotations. An example of such a calibration measurement is shown in Extended Data Fig. 4k.

**CZ phase crosstalk (Q1, Q2, Q5, Q6→<30 s; Q3, Q4→<50 s).** After the exchange pulse is executed, single-qubit phases have to be corrected.

We correct these phases on all the qubits, whether participating or not in the two-qubit gate. We calibrate the required phase corrections in a very similar way as done for the single-qubit gate phase corrections. An example of the circuit and measurement is given in Extended Data Fig. 4l,m. The exact calibration run-time depends on the CZ pulse width and can vary by a couple of seconds depending on the target qubit.

## Heating effects

We observed several effects that bear a signature of heating in our experiments. When microwaves are applied to the EDSR line of the sample, several qubit properties change by an amount that depends on the applied driving power and the duty cycle of applying power versus no power. This effect has also been observed in other works[55]. We report our findings in Extended Data Fig. 6 and will discuss adjustments made to the sequences of the experiments to reduce their effects. The main heating effects are a reduction of the signal-to-noise ratio (SNR) of the sensing dot and a change of the qubit resonance frequency and $T_2^*$.

In Extended Data Fig. 6a–d, we investigate the effect of a microwave burst applied to the EDSR driving gate, after which the signal of the sensing dot is measured. We observe changes in the background signal and in the peak signal (the electrochemical potential of the sensing dot is not affected, as the peak does not shift in gate voltage). As the background signal rises more than the peak signal, the net signal is reduced. This reduction depends on the magnitude and duration of the applied microwave pulse (Extended Data Fig. 6b). The original SNR can be recovered by introducing a waiting time after the microwave pulse. The typical timescale needed to restore the SNR is of the order of 100 μs (Extended Data Fig. 6c,d). We added for all (randomized benchmarking) data taken in this paper a waiting of 100 μs (500 μs) after the manipulation stage to achieve a good balance between SNR and experiment duration. Spin relaxation between manipulation and readout is negligible, given that no $T_1$ decay was observed on a timescale of 1 ms within the measurement accuracy. We did not introduce extra waiting times after feedback/CROT pulses in the initialization/readout cycle, as the power to perform these pulses did not limit the SNR.

Extended Data Figure 6f gives more insight in what makes the background and peak signal of the sensing dots change. The impedance of the sensing dot is measured using RF reflectometry. The background of the measured signal depends on the inductance of the surface-mount inductor, the capacitance to ground[29,56,57] and the resistance to ground of the RF readout circuit. Extended Data Figure 6f shows the response of the readout circuit under different microwave powers (the RF power is kept fixed). A frequency shift (0.5 MHz) and a reduction in quality factor is observed. This can be indicative of an increase in capacitance and dissipation in the readout circuit. Currently the microscopic mechanisms that cause this behaviour are unknown.

The second effect is observed when looking at the qubit properties themselves. Extended Data Figure 6e shows that both the dephasing time $T_2^*$ measured in a Ramsey experiment and the qubit frequency are altered by the microwave radiation. In the actual experiments, we apply a microwave pre-pulse of 1–4 μs before the manipulation stage to make the qubit frequency more predictable, although this comes at the cost of a reduced $T_2^*$. The pre-pulse can be applied either at the start or at the end of the pulse sequence, with similar effects. This indicates that heating effects on the qubit frequency persist for longer than the total time of a single-shot experiment (approximately 600 μs), which is different from the effect on the sensing dot signal. Also the microscopic mechanisms behind the qubit frequency shift and $T_2^*$ reduction remain to be understood.

## Parity mode PSB readout

PSB readout is a method used to convert a spin state to a more easily detectable charge state[58]. Several factors need to be taken into account for this conversion, to enable good readout visibilities. Extended Data

Figure 1a,b shows the energy level diagrams for PSB readout performed for (1,1) and (3,1) charge occupation. The diagrams use valley energies $E_v$ of 65 µeV to illustrate where problems can occur. When looking at Extended Data Fig. 1, we can observe two potential issues:

- The excited valley state with $|\downarrow\downarrow\rangle$ is located below the ground valley state with $|\uparrow\downarrow\rangle$. We assume in the diagram that the (2,0) singlet state ($|S,0\rangle$) is coupled to both the (1,1) ground valley state and the (1,1) excited valley state. In this case, during the initialization/readout pulses, a population can be moved into the excited valley state. This problem can be solved by working at a lower magnetic field, such that $E_v > E_z$ (Extended Data Fig. 1b).
- When operating in the (1,1) charge occupation, the readout window is quite small, as the size is determined by the difference between the valley energy and the Zeeman energy. A common way to prevent this problem is by operating in the (3,1) electron occupation.

With both measures in place, we consistently obtain high visibilities of Rabi oscillations (≥94%) on every device tested.

In the following we describe the procedure used to tune up the parity mode PSB.

- Find an appropriate tunnelling rate at the (3,1) anticrossing. An initial guess of a good tunnelling rate can be found using video mode tuning. We use the arbitrary waveform generator to record at high speed the frames of the charge stability diagram (5 µs averaging per point, a full image is acquired in $t_{image} = 200$ ms). During the measurement of the frames, we vary the tunnel coupling, while looking at the (3,1) ↔ (4,0) anticrossing until the pattern shown in Extended Data Fig. 1c is observed. This figure shows that, depending on the (random) initial state, the transition from (3,1) to (4,0) occurs at either location (i) or location (ii). This is exactly what needs needs to happen when the readout is performed.
- Find the readout point. We hold point (1) fixed in the centre of the (3,1) charge occupation (Extended Data Fig. 1c). Point (2) is scanned with the AWG along the detuning axis as shown in Extended Data Fig. 1c. We pulse from point (1) to point (2) and measure the state (with ramp time of around 2 µs), and then we pulse back to point (1). When plotting the measured singlet probability, a gap is seen between the case where a singlet is prepared and the case where a random spin state is prepared (Extended Data Fig. 1d). The centre of this region is a good readout point.
- Optimizing the readout parameters. The main optimization parameters are the detuning ($\epsilon$), tunnel coupling ($t_c$) and ramp time to ramp towards the PSB region. We also independently calibrate the ramp time and tunnel coupling from the readout zone towards the operation point of the qubits. When ramping in towards the readout point, it is important to be adiabatic with respect to the tunnel coupling. We do not need to be adiabatic with respect to spin, as both $|\uparrow\downarrow\rangle$ and $|\downarrow\uparrow\rangle$ relax quickly to the singlet state (faster than we can measure, in less than 1 ns). When pulsing from the readout to the operation point, more care has to be taken. When using the ramp time that performs well for the readout, we notice that we initialize a mixed state, as we are not adiabatic with respect to spin. This can be solved by pulsing the tunnel coupling to a larger value before initiating the initialization ramp (Extended Data Fig. 1g).

We show in Extended Data Fig. 1e,f that the histograms for parallel and antiparallel spin states are well separated, which enables a spin readout fidelity exceeding 99.97% for both qubits Q1–Q2 and for qubits Q5–Q6. This number could be further increased by integrating the signal for longer, but is not the limiting process. This method of quantifying the spin readout fidelity is commonly used in the literature but it leaves out errors occurring during the ramp time (the mapping of qubit states to the readout basis states). This can be a pronounced effect, as seen from the measured visibility of the Rabi oscillations.

**Postselection of data.** When using parity readout on a single qubit pair, around 5% of the runs are discarded on average as part of the

initialization procedure (Fig. 1d). In the case when two outer qubit pairs are used, about 10% of the data are discarded ($1 - 0.95^2$). When performing experiments on all six qubits, additional initialization steps with postselection are needed (in Extended Data Fig. 2, runs are postselected on the basis of 18 measurement outcomes in total), and we discard around 65% of the dataset. When we do not discard any runs, the initialization fidelity reduces by around 5–9% for a single qubit pair.

### Setup and the real-time feedback using FPGA

**Setup.** A detailed schematic of the experimental setup is presented in Extended Data Fig. 7, listing all the key components used in the experiment.

**Programming quantum circuits.** The quantum circuits are implemented in the form of microwave bursts for single-qubit operations, gate voltage pulses for two-qubit gates and gate voltage pulses combined with RF bursts for readout. The gate voltage pulses are generated by an arbitrary wave generator (AWG). The microwave bursts are generated through IQ modulation of a microwave vector source carrier frequency. The input signals for the IQ modulation are generated by the same AWG as used for the voltage pulses. The IQ modulation defines the amplitude envelope of the microwave bursts, the output frequency and the phase shifts. Virtual $Z$ gates are implemented by incrementing the reference phase of the numerically controlled oscillators (NCOs) (see below) and are used to, for example, correct phase errors introduced by crosstalk. The generated control signals are stored in memory with a resolution of 1 ns.

Microwave bursts applied to the six-qubit sample are supplied by a single microwave source with a carrier frequency set at 16.3 GHz. We address the six different qubits using single side-band IQ modulation of the carrier to displace the frequency of the microwave output signal to the frequency of the target qubit. As each qubit has a different resonance frequency (which is different from the carrier frequency), it is necessary to track the phase evolution at the qubit Larmor precession frequency to ensure phase coherent microwave bursts for successive single-qubit operations. To realize that, we define in the AWG six continuously running NCOs, one for each qubit. These NCOs keep track of the phase evolution of the qubits with respect to the carrier frequency. We choose this approach instead of precalculating the phase factors for every pulse in a sequence, which is a not a scalable approach with the growing complexity of the quantum circuits.

The digitizer is synchronized with the AWG to acquire qubit readout data. In a single shot we can include multiple readout segments, each defined in a digitizer instruction list. A step in this list specifies a measurement time window, a wait time and the threshold for the qubit state. The input signal is integrated during the measurement window and the result is compared with a threshold to determine the qubit state. This outcome, 0 or 1, can be passed directly to the AWG by a trigger line within a Keysight PCI eXtensions for Instrumentation (PXI) chassis, shared by the digitizer and the AWGs, to realize real-time feedback on the measurement output.

**Real-time feedback.** In the initialization and readout sequences the execution of selected gates depends on the outcomes of intermediate measurements, which enables real-time qubit state corrections. The total time from the end of the measurement until the start of the conditional gate (burst) on the device should be much shorter than the qubit relaxation time $T_1$, and ideally also shorter than around 1 µs, which is the time needed for the adiabatic passage back to the manipulation point after the parity measurement, such that no unnecessary idling time is spent. This fast control loop is realized with a custom FPGA (field-programmable gate array) program in the AWG and digitizer as shown in Extended Data Fig. 8. The total latency for the closed loop feedback is 660 ns, which fits the design requirements.

## Data availability

The raw data and analysis that support the findings of this study are available in the Zenodo repository (https://doi.org/10.5281/zenodo.6138474)

## Code availability

The measurement and analysis code is available in the Zenodo repositories (core-tools https://zenodo.org/badge/latestdoi/264858832; pulse library https://zenodo.org/badge/latestdoi/113251242; qubit abstraction layer https://zenodo.org/badge/latestdoi/253903530; state tomography https://zenodo.org/record/6135943).

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

**Acknowledgements** We acknowledge R. Schouten for general advice and help on the measurement electronics, M. Almendros and his team for a collaborative development of FPGA hardware control, H. Van Der does and N. Philips for the design of the sample printed circuit board, Z. Jiang, L. Peters and F. Unseld for assistance with the testing of samples, M. Eriksson and his team for contributions to sample fabrication, and members of the Vandersypen group for useful discussions. We acknowledge financial support from the Marie Skłodowska-Curie actions-Nanoscale solid-state spin systems in emerging quantum technologies-Spin-NANO, grant agreement number 676108. This research was sponsored by the Army Research Office (ARO) under grant numbers W911NF-17-1-0274 and W911NF-12-1-0607. The views and conclusions contained in this document are those of the authors and should not be interpreted as representing the official policies, either expressed or implied, of the ARO or the US Government. The US Government is authorized to reproduce and distribute reprints for government purposes notwithstanding any copyright notation herein. Development and maintenance of the growth facilities used for fabricating samples is supported by DOE (DE-FG02-03ER46028). We acknowledge support from Keysight's University Research Collaborations.

**Author contributions** S.G.J.P. and M.T.M. performed the experiment with help from C.V. Data analysis was carried out by S.G.J.P., M.T.M. and M.R., who also performed the numerical simulations of the Bell and GHZ states. S.G.J.P. and S.L.S. wrote the libraries used to control the experiment. S.L.S. wrote the library used for real-time feedback and made the supporting FPGA images. S.G.J.P., M.T.M., M.R. and L.M.K.V. contributed to the interpretation of the data. S.V.A., N.K., D.B., W.I.L.L., M.V. and L.T contributed to device fabrication. A.S., B.P.W. and G.S designed and grew the Si/SiGe heterostructure. S.G.J.P., M.T.M. and L.M.K.V. wrote the manuscript with comments by all authors. L.M.K.V. conceived and supervised the project.

**Competing interests** The authors declare no competing interests.

**Additional information**
**Correspondence and requests for materials** should be addressed to Lieven M. K. Vandersypen.

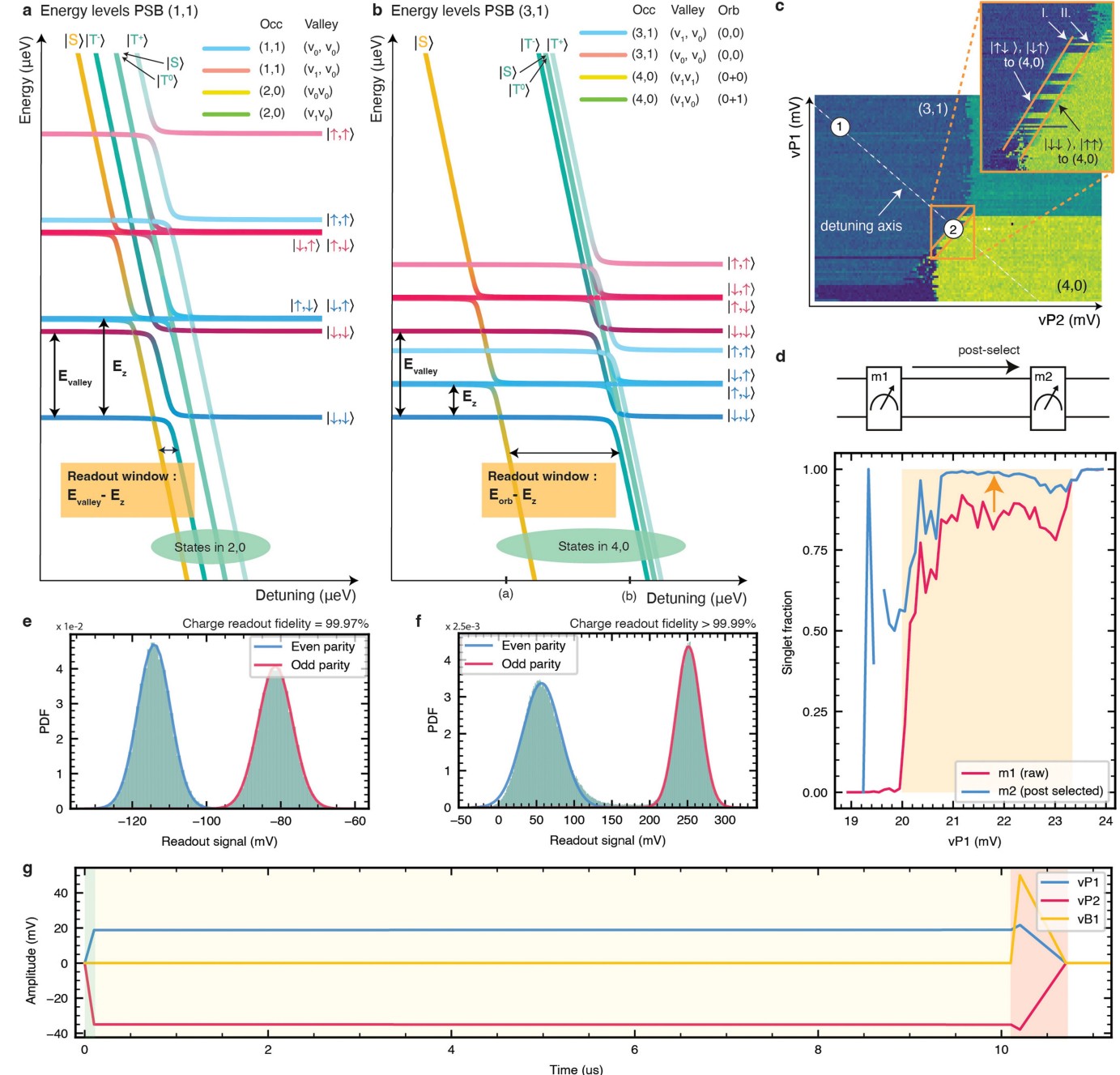

**Extended Data Fig. 1 | Pauli Spin Blockade readout. a**, Energy diagram for a double quantum dot as a function of the detuning between the (1,1)/(2,0) charge electron occupation. The Zeeman energy ($E_z$) for qubit Q1, Q2 is set to 74, 75 $\mu$eV (18/18.2 GHz) and the valley energy ($\epsilon_v$) of dot 1 is set to 65 $\mu$eV. We set $\epsilon_v$ for the second dot to a much larger value to shift part of the energy spectrum out of view and simplify the visual analysis. The charge and valley occupations are indicated in the top right of this panel. **b**, Energy diagram in the (3,1)/(4,0) charge occupation. This panel uses identical parameter values as panel **a**, except for the Zeeman energy for qubit Q1 and Q2, 25 and 26 $\mu$eV (6 and 6.2 GHz). The excited state energy of dot 1 in the (4,0) charge occupation is given by the orbital energy instead of the valley splitting. **c** Experimental charge stability diagram taken at the (3,1)/(4,0) anticrossing for dots 1 and 2. The point indicated with '1' indicates the qubit operation point and the point indicated as '2' indicates the readout point. The inset zooms in on the anticrossing, allowing one to observe the spin selective tunneling for the different input states (the readout zone). **d** In our experiment we initialize via measurement and

post-selection. In the plot we can see the effects of two subsequent readouts. First readout 'm1 (raw)' shows the initial singlet fraction (electron spins are not intentionally randomized and by nature of the executed measurement sequence, a singlet state is preserved for next single shot). Second measurement 'm2' shows the outcome from post-selection on the result of 'm1' (realized as per-measurement-point post-processing in software). Within the readout zone (shaded area) the initialized singlet fraction is greatly amplified. **e**–**f** Probability density function of the PSB readout signal between qubit pairs Q12 and Q56 (10 $\mu$s integration time), recorded in the course of the Rabi oscillations of qubit Q1, Q5 in figure 2. The Gaussians are fitted to the two distributions and charge readout fidelities, estimated from their overlap, exceeding 99.9%. No $T_1$ decay was observed at the readout point ($T_1 >> 100 \mu$s). **g**, Gate voltage pulses applied to perform PSB readout on qubit pair Q12. The different background colors indicate the ramp towards the readout point (green), measurement at the readout point (yellow) and the adiabatic ramp back to the operating point (red).

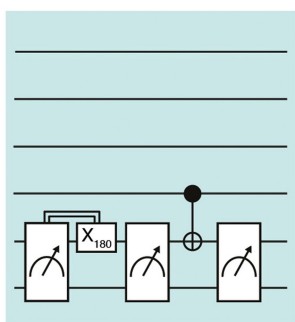

**a**

**Initialization**

**Q234 GHZ-state preparation**

**Readout**

**Extended Data Fig. 2 | Full quantum circuits. a**, This circuit was used to produce the data presented in Fig. 5c. The different background colors indicate the different parts of the sequence (yellow – initialization, green – manipulation (including the tomography pulses), blue – readout). The sequence could be made much shorter if we would parallelize the readout operations. We did not implement this, since this is not the limiting factor ($t_{read} << T_1$).

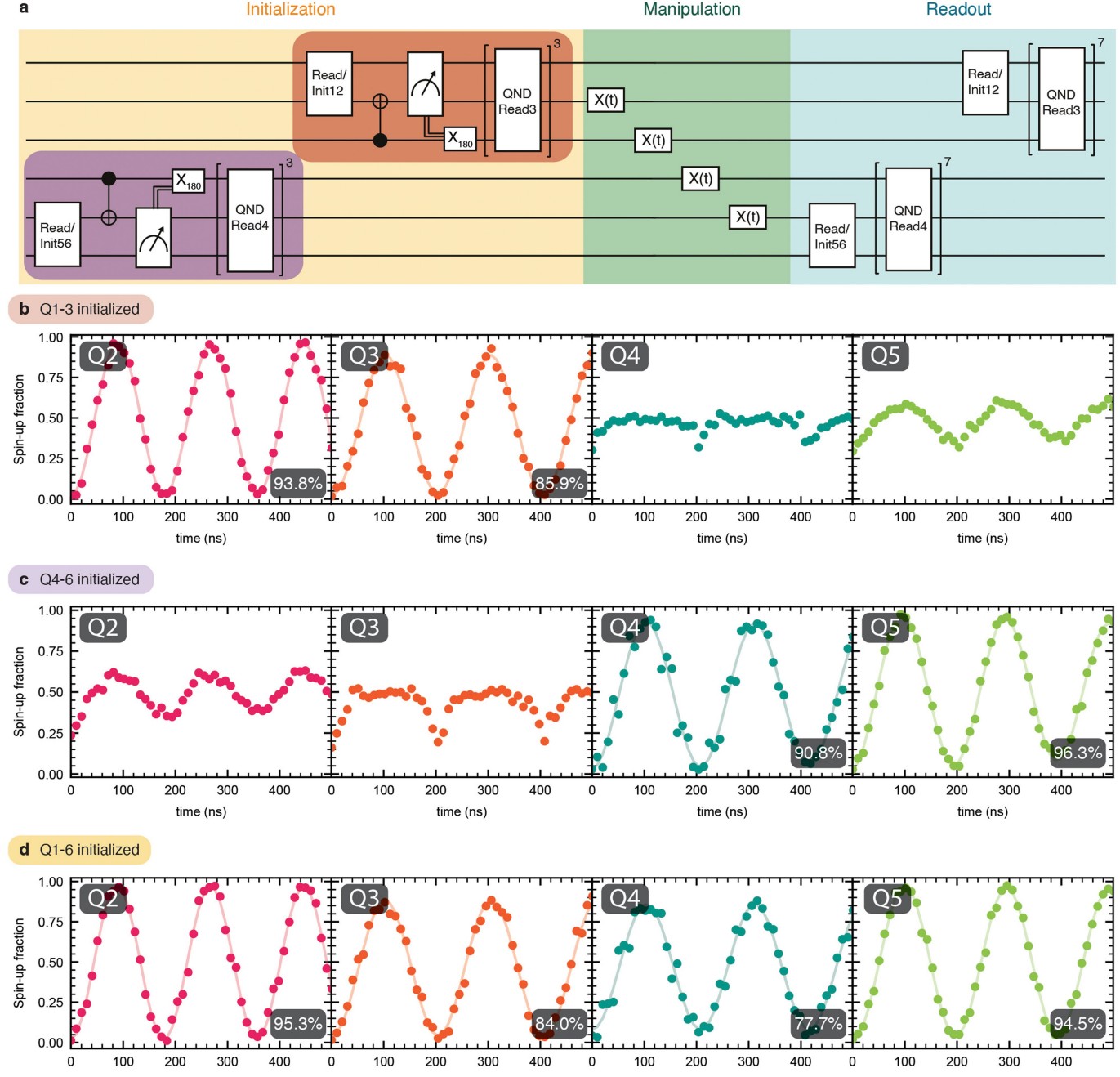

**Extended Data Fig. 3 | Loss in visibility when initializing all qubits. a**, Gate sequence used to demonstrate the effect of running different qubit initialization routines. Microwave bursts with variable duration are performed sequentially on qubits Q2, Q3, Q4 and Q5, as shown in the schematic. In all cases, readout is performed on all the qubits, but we vary which qubits are initialized for a particular experiment. **b**–**d**, Results of the sequence displayed in panel **a**, when qubit Q1-Q3 (**b**), Q4-Q6 (**c**) or Q1-Q6 (**d**) are initialized. The shaded numbers in the panels indicate the visibility of the measured qubit. With all qubits initialized we observe a visibility loss on qubit Q4 and to a lesser extent qubit Q3. A visibility loss in principle can originate from a reduced readout or initialization fidelity. We keep the readout sequence identical for

**a, b**, and **c**. Additionally, we include a $500\,\mu s$ waiting time prior to readout, to minimize any effects of MW pulsing during initialization or manipulation stage on the readout performance (see Methods). Although, we cannot be certain that the readout fidelity is unaffected by the initialization of all qubits, we speculate that the majority of the observed visibility loss is due to a reduced initialization fidelity, possibly due to the sensitivity of the CROTs to qubit frequency shifts. This interpretation is consistent with the fact that mostly qubit Q4 suffers a lower visibility, as qubit Q4 is initialized before qubits Q1, Q2 and Q3. If instead we reverse the initialization order, qubit Q3 displays lowered visibility (data not shown).

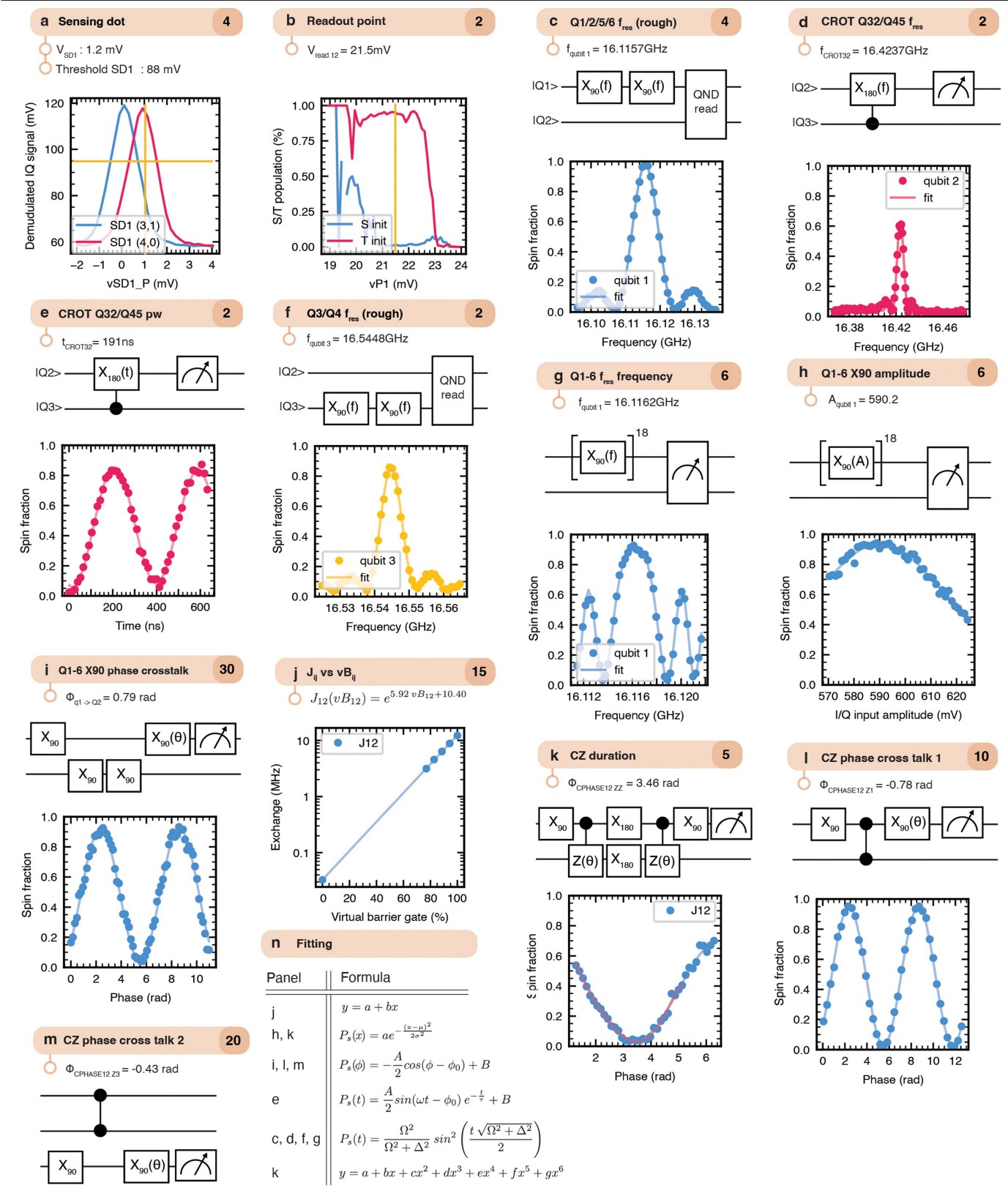

**Extended Data Fig. 4 | Calibrations. a–m** Each panel shows a typical experimental dataset obtained in one of the calibration routines for the relevant experimental parameters. Every panel indicates in the header the calibration name and the corresponding number of calibration parameters.

Below the header, the values extracted from the data is indicated. See the methods section for a detailed description. **n**, Table showing the formulas used to fit the data for the indicated figure panels.

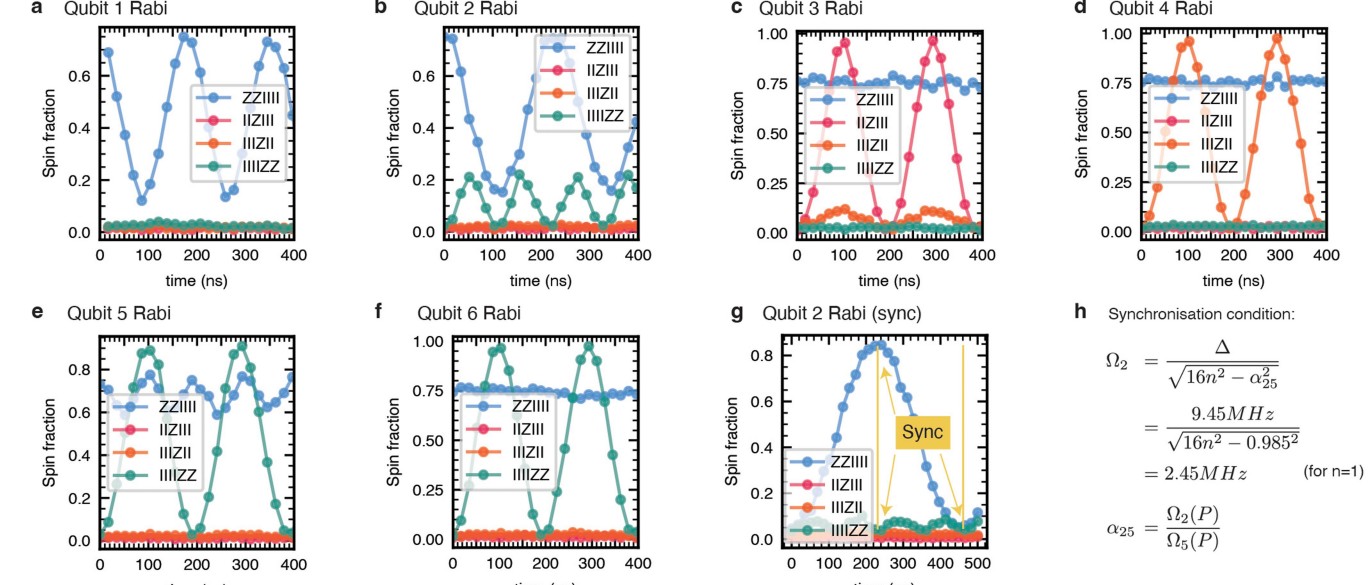

**Extended Data Fig. 5 | Crosstalk. a–f,** In this experiment all qubits are initialized. We observed the presence of crosstalk by measuring the expectation values of the native observables on the sample. **a–e,** Rabi oscillations executed on different qubits in each panel. Panels b, c and e display coherent oscillations of nominally idle qubits, which are a clear indication of crosstalk at the chosen Rabi frequencies. Qubits Q1 and Q2 show flipped measurement outcomes due to a miscalibration of sensing dot 1. This bears no influence on the conclusion from this experiment. **g,** Example of qubit Q2 driven at the frequency determined by the synchronization condition. In this case the crosstalk to qubit Q5 (qubit with the closest resonance frequency) was nullified by design for every multiple of a 90 degree rotation. **h,** Expressions[53] used to calculate the synchronization condition shown in panel **g.**

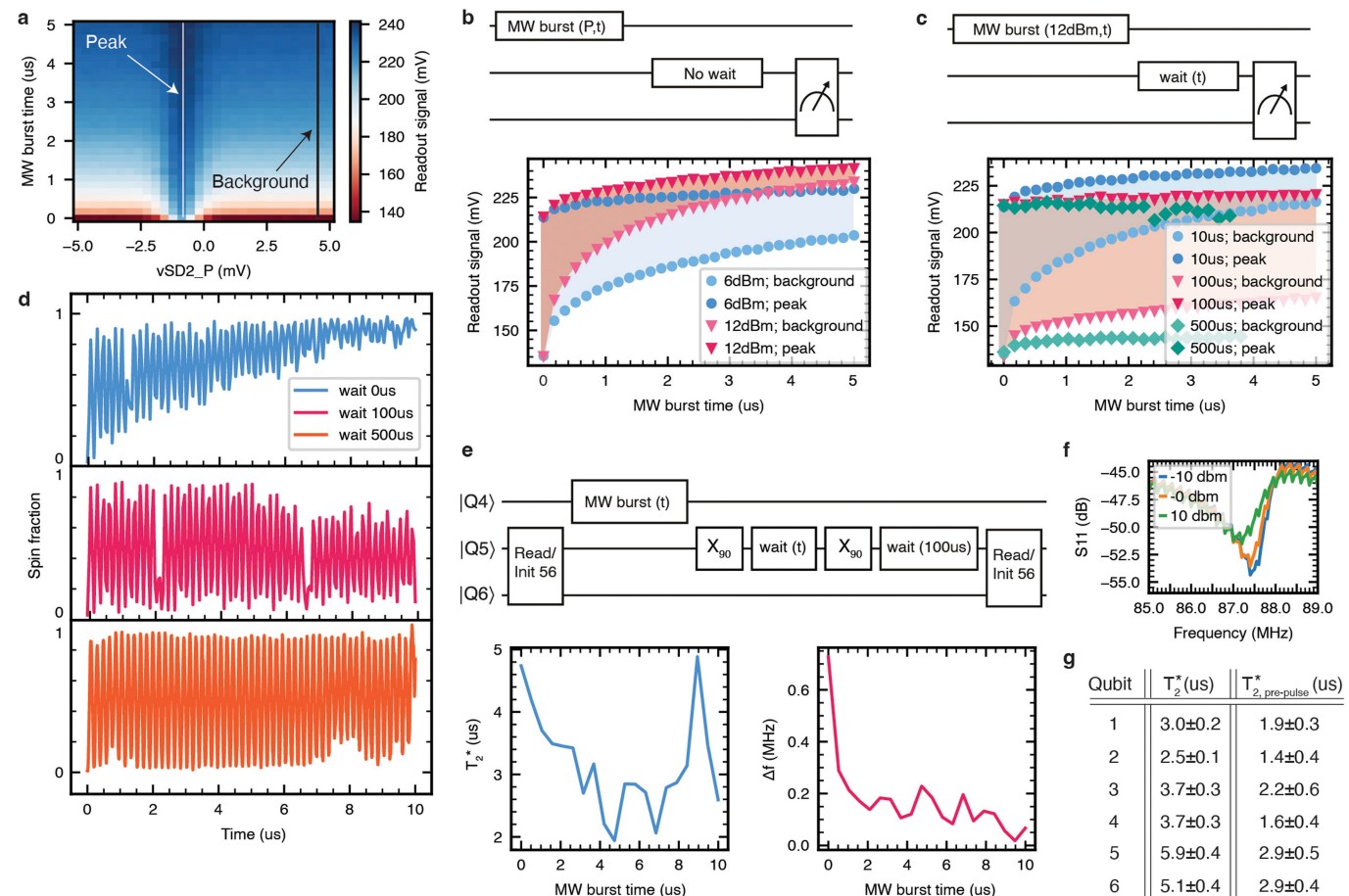

**Extended Data Fig. 6 | Heating effects. a**, A microwave burst of 16 GHz at 12 dBm output power and of a variable duration is applied before scanning the sensing dot virtual plunger gate. We observe no shift of the Coulomb peak, indicated by the white line, but the background signal increases substantially. Also the peak signal increases, but by a smaller amount. **b**, Linecut along the Coulomb peak and linecut parallel to the Coulomb peak of panel **a**, showing the peak and background signal as a function of the MW burst duration, for two different powers. The shaded area indicates the net signal, which is smaller the larger the applied power and the longer its duration. **c**, Variation on panel **b**, where we introduce a waiting time after the microwave burst, as indicated in the schematic. The longer the wait time, the more the original SNR is recovered, and the higher the readout fidelity will be. **d**, Rabi oscillations with different waiting times introduced before the qubit readout. The 500 μs wait time allows for recording long lived Rabi oscillations, while with no wait time prior readout, the contrast vanishes as the perceived spin fraction converge towards 1, due to shifts in the sensing dot signal and background (the threshold for single-shot analysis was kept fixed). **e**, Schematic showing the circuit used to investigate the effect of a pre-pulse (labelled MW burst (*t*)) on the qubit properties. A microwave burst of 6 dBm is applied before running a Ramsey experiment. We extract the change in $T_2^*$ and Larmor frequency for qubit Q5 for different microwave burst times as shown in the plots. **f** Return loss of the RF readout circuit for different powers of continued microwave driving (i.e. driving at the qubit frequencies, not for the RF readout). We observe both a shift in the RF resonance frequency and a degradation of the quality factor with higher power excitation. **g**, Extracted dephasing times $T_2^*$ with and without pre-pulse (4 μs, 6 dBm), measured as illustrated in **e**.

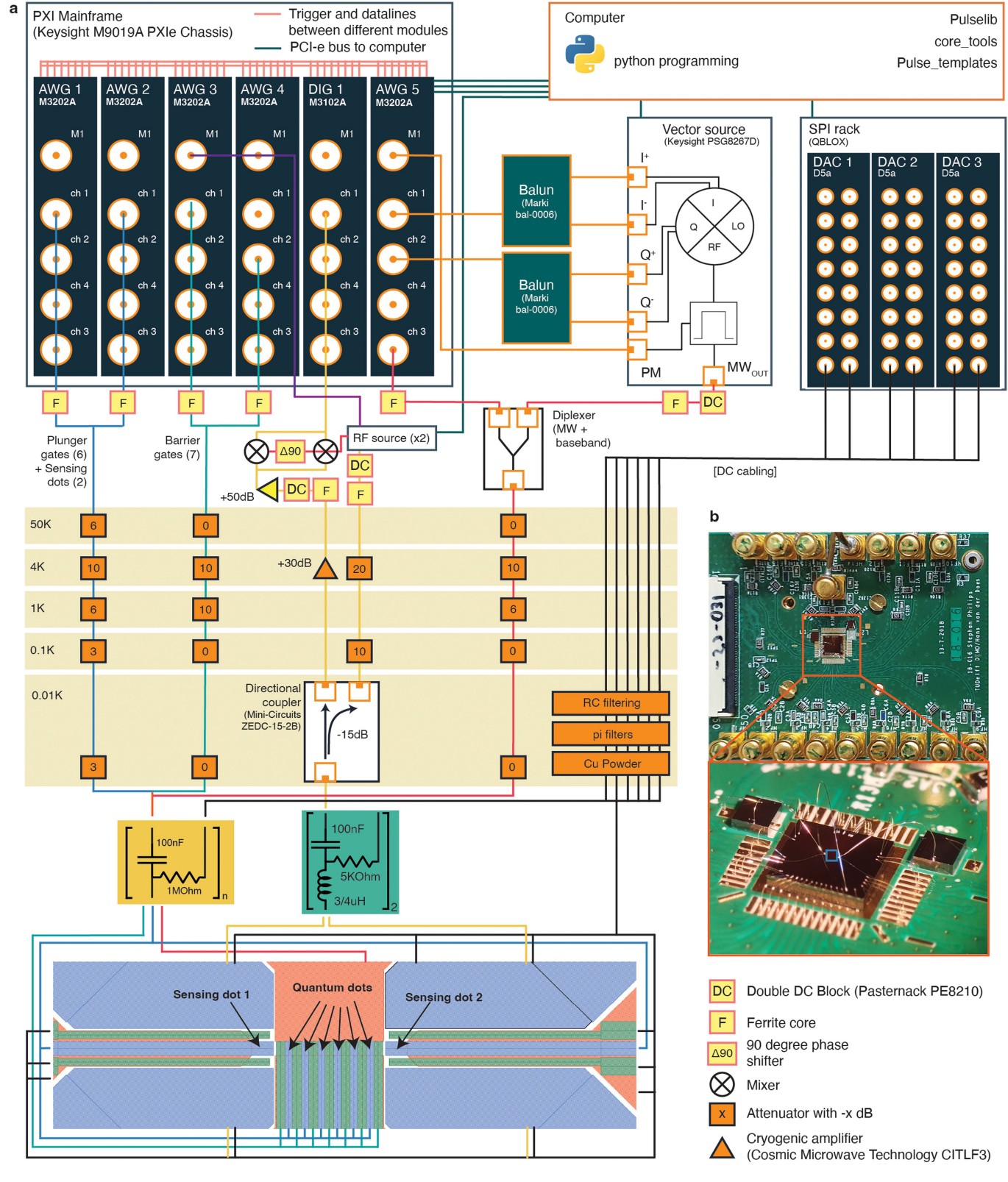

**Extended Data Fig. 7** | See next page for caption.

**Extended Data Fig. 7 | Experiment setup. a**, Schematic overview of the experimental setup. The bulk of the experiment is controlled by Arbitrary Waveform Generators (AWG, Keysight M3202A) and digitizers (DIG, Keysight M3201A) in a PXI chassis (used for synchronization and feedback). The AWG's generate baseband pulses (0-300MHz) used for readout and two-qubit gates. These baseband signals are provided to all plunger and barrier gates of the quantum dots. Sensing dot plungers are also connected to AWG channels to allow for fully compensated virtual plungers. In addition, we use the AWG's to generate the IQ input signals for the vector source (Keysight PSG E8267D), used to perform single-qubit gates. Using this source, a large IQ modulation bandwidth (800 MHz) can be obtained by using the differential IQ inputs. The differential signal is generated using Balun's (Marki bal-0006), which reduce the number of AWG channels and ensure excellent timing (AWG channel pairs 1,2 and 3,4 have a larger skew ( ± 30 ps) compared to just channel 1 and 2). We use a homemade combiner to allow for both baseband and MW control on the EDSR driving gate. Coils with ferrite cores are used to reduce low-frequency noise generated by the instruments. In addition, we use double DC blocks for any RF/MW signal used in this experiment. The schematic shows in yellow the different temperature stages of the dilution refrigerator at which the signals are attenuated and thermalized. All plunger gates have discrete attenuators in the line with a total attenuation of - 28 dB, and barrier gates of - 20 dB in addition to the attenuation from the coax line itself. The barriers gates have less attenuation because of the large voltage pulses needed to achieve the desired $J_{on}/J_{off}$ ratio. We use bias tees on the sample PCB with a RC time constant of 100 ms to combine baseband and microwave signals with a static DC voltage. We generate the carrier signal using a homemade RF source (one carrier per sensing dot), which we route into the dilution refrigerator using a combiner. A marker output channel of the AWG's is connected to the the RF sources in order to only output RF power during the readout. At the 10 mK stage, we use a directional coupler to separate the reflected signal ($S_{11}$) from the incoming RF carrier. A coplanar waveguide routing the RF signal on the sample PCB is split in two and connects to bias tees, each one going to a sensing dot. Bias tees with a low resistance are chosen as it also allows us to perform DC measurements as needed. NbTiN inductors (low $C_p$, high Q) are wire-bonded directly to the source contacts on the sample[29,56] (see Supplementary Information). When the carrier signal reflects from the sensing dots, it passes again through the directional coupler and is amplified both at the 4 Kelvin stage and at room temperature. The signal is fed through two sets of mixers (1 for each SD) to demodulate the signal to baseband. We finally sample the I and Q channels with the digitizer in the PXI chassis. On the FPGA, we average the signal for a specified amount of time and optionally convert it into a boolean value using a threshold (see Extended Data figure 8). Besides the fast baseband/RF/MW pulses, all the gates of the sample are also connected to battery-powered DACs built in-house, which supply the DC operating voltages. These DACs are very stable voltage sources that provide an 18-bit voltage resolution over a ± 2 V range. **b**, Image of the PCB used to mount the sample. The DC signals are supplied via the white flat-flexible-cable connector located on the left side of the PCB. The 17 SMP connectors provide the signals used for qubit readout and single- and two-qubit gates. We use bias tees to combine the AC and DC signals. A laser diode is placed in the top right corner of the PCB, which can be used to 'reset' the device (not used in this experiment). In the zoomed-in image in red, the qubit chip and two smaller chips with NbTiN high-kinetic inductance inductors are visible.

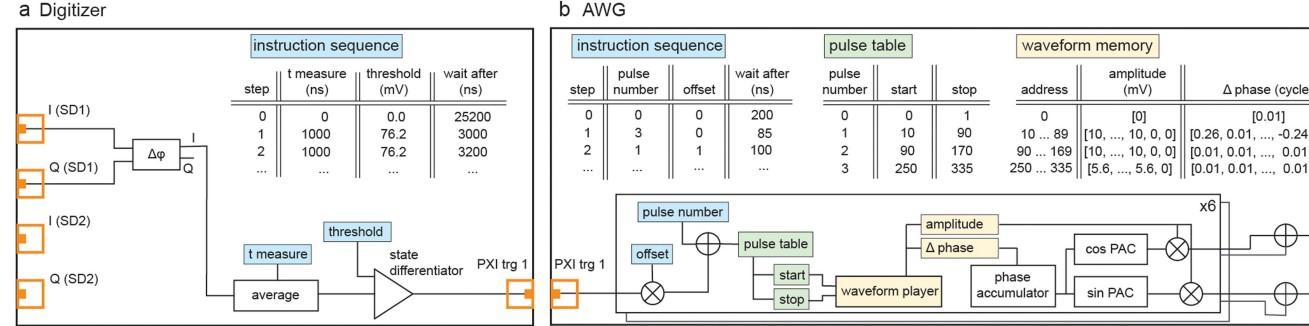

**Extended Data Fig. 8 | Implementation of real time feedback. a**, The sensing dot signal obtained via RF reflectometry arrives at the digitizer on two input channels (I and Q). The digitizer rotates the combined I and Q input with an angle $\Delta\phi$ and converts the vector into a scalar by dropping the Q signal. Upon a trigger from the instruction sequence, the signal is averaged for a time $t_{measure}$ and compared with a threshold to infer the qubit state. The result is written both to the DRAM and the PXI trigger line. **b**, We use IQ modulation to shape the MW pulses that are used for EDSR qubit control. The waveform memory stores the amplitude (envelope) and phase information for all the microwave bursts used in the experiment, as well as for the necessary single-qubit phase corrections. We upload waveforms for microwave bursts corresponding to $X_{90}$, $X_{-90}$, $Y_{90}$ and $Y_{-90}$ rotations for each qubit. The pulse table contains the start and stop memory addresses of each control pulse present in the waveform memory. For every single-shot experiment, the AWG steps through the instruction sequence, which defines all the single-qubit gate pulses that need to be executed during the experiment. When the offset flag of an instruction in the instruction sequence is 1, the current value on the PXI trigger (0/1) is added to the pulse number that will be played from the pulse table. This bitwise addition implements real-time feedback. In the present experiment, either $X_{90}$ bursts will be executed or zero amplitude bursts, depending on the PXI trigger value. When an instruction sequence is ran, the amplitude and phase information are read from the waveform memory for the selected pulse numbers one after the other. The differential phase ($\Delta$ phase) is added for every rendered sample to the phase accumulator (which controls the qubit frequency) and is then converted to an in-phase (I) and quadrature (Q) signal by the phase-to-amplitude converters (PAC). These I and Q signals are multiplied with the amplitude envelope of the waveform and are then passed to the outputs of the AWG and from there to the vector source. We can run up to twelve sequencers in parallel in a single AWG. In this case 6 sequencers are used, one for every qubit.

**Extended Data Table 1 | Exchange pulses**

|         | vB0 | vB1  | vB2  | vB3  | vB4  | vB5  | vB6 |
|---------|-----|------|------|------|------|------|-----|
| $V_{J_{12}}$ | -1  | 1    | -1.6 | 0    | 0    | 0    | 0   |
| $V_{J_{23}}$ | 0   | -0.2 | 1    | -0.5 | 0    | 0    | 0   |
| $V_{J_{34}}$ | 0   | 0    | -0.3 | 1    | -0.3 | 0    | 0   |
| $V_{J_{45}}$ | 0   | 0    | 0    | -0.9 | 1    | -0.9 | 0   |
| $V_{J_{56}}$ | 0   | 0    | 0    | -0.2 | -0.9 | 1    | 0   |

In order to achieve sufficiently high $J_{on}/J_{off}$, we use a combination of barrier gate pulses, where we pulse a barrier gate in between the target qubit pair to a more positive voltage and at the same time we pulse the voltage on the barrier gates on the outer side of target qubit pair to a less positive voltage. This pushes the two quantum dots towards each other further enhancing the tunnel coupling.