## [Peer Review File · Nature]

Manuscript Title: Universal control of a six-qubit quantum processor in silicon

Reviewer Comments & Author Rebuttals

Reviewer Reports on the Initial Version:

Referees' comments:

Referee #1 (Remarks to the Author):

Review of the manuscript "Universal control of a six qubit quantum processor in silicon" by Stephan G. J. Philips et al., submitted to Nature.

In this work, the authors demonstrate the first semiconductor 6-qubit chip. For the first time, they combine single- and two-qubit gates with high-fidelity initialization and read-out strategies. Their demonstration sets a milestone for semiconductor spin qubits. The authors discuss how to operate and tune their dense linear array of qubits and point out that 108 parameters have to be tuned. The authors take many efforts to provide all details about this new large-scale tuning efforts and the related time-budget (methods and extended data Fig. 4 and in supplementary materials).

Uncertainties are treated very well and data is robust. The applied methods are not new and the fidelities are not the highest compared to previous publications which just focus on one functional block or fidelities measured on small demonstrators (with smaller qubit number), but this cannot be expected when combining all functionalities in one multi-qubit device for the first time. The resonant frequency of the qubits did not turned out be as planned (the authors provide a good analysis in the supplementary material). Nevertheless, the authors could compensate this and manage to make the qubits nearly "identical" in terms of Rabi frequency and exchange interaction.

The manuscript is very clearly written and exemplary for transparency on all methods. The presentation quality is very high. In fact, it is amazing how many details about experimental setup, the sample design, heating effects, cross-coupling, sample evolution are provided by the authors. This paper is a highlight on spin-qubit quantum computing and one of the best I ever read. It is definitely recommend publication in Nature.

Some minor suggestions/questions:

96: "Figure 1d" has to be changed to "Figure 1c".

151: "Figure 2c" has to be changed to "Figure 2b".

Caption Fig. 2: "qubit frequency for each the six qubits" change to "qubit frequency for each OF the six qubits"

Caption Fig. 2: I suggest a more suitable title: "Experiment overview" could be replaced by "Single qubit gate characterization"

Colours are hard to distinguish in Fig. 1d.

Fig. 1a is flipped compared to Extended Data Fig. 7

215: "Twice a week, we run...". This information is repeated in the Method section.

1. Extended Data Fig. 7 shows a laser diode. Was it required for the experiment?
2. Is it correct that the Co micromagnets are not covered in order to prevent oxidization?
3. Does the 3-electron filling of the outer QD alter the expectation for the resonant frequency due to e.g. different g-factor?
4. Conclusion: For such an exemplary paper, the authors might elaborate more on (their opinion) how to scale-up spin qubit chips in the future. What are the main upcoming challenges? What has to be developed next?

Referee #2 (Remarks to the Author):

The manuscript by Philips et al. reports on universal control achieved over a linear array of six quantum dots, forming six Loss-DiVincenzo spin qubits. The experiments performed are extremely technically challenging, and it is a feat to reliably perform the voltage pulses and calibrations required to operate these complex quantum gates in a multi-dimensional gate space. The manuscript is very well written, and I congratulate the authors on getting all aspect of this difficult experiment to work. In fact, the main strength of the paper is in combining state-of-the-art initialisation, readout, and operation fidelities with consistent fabrication, FPGA-based improvements such as the active reset, and keeping all of this consistent while scaling to six quantum dots. In all, I think the current demonstration is a large step forward for the field of spin qubits and solid-state qubits. While it also posits a lot of questions (regarding maintenance of coherence, gate fidelity as the number of qubits is scaled, and design and routing scaling into the second and third dimension), these are topics crucial to explore and therefore the current work will open more areas of further research in the field, and should be brought to the notice of the wider community. Therefore I would recommend publication in Nature.

I have some comments and questions I would like the authors to address:

- 1) The authors mention that they have solved the previous problem of low valley splittings, but I could not find an explanation how. Could they comment on this? One of the devices measured still suffered from this issue, so it seems to be still relevant.
- 2) The caption of Fig. 1 contains the text "...up to spin relaxation"; what is this value of overall spin relaxation time, especially for the outer spins? I could not find a reference to this, only summer lower bounds. Does it vary across the array? Also, in Fig. 1e, the spin symbols and their colour are unclear

at first glance and could be explained in the caption. Also, this section should refer to Extended Data Figure 2 (this is currently referred to only in the text) as the caption should be standalone, and it is hard to parse without referring to EDF2.

3) The parity readout method is clever in solving the problem of the sensors being insensitive to the inner dots, but post-selection means many shots could be disregarded. What are the implications of this for the operation of the processor and how much overhead does this add to the operation? How many runs are typically discarded?

4) The paper in generally performs readout and operation sequentially. The authors should clearly explain why this is not performed simultaneous? (I assume frequency multiplexing is possible as this is used for the EDSR antenna.) Is this limited by readout crosstalk? For operation, the fact that crosstalk is an issue is similarly buried in Extended Data Figure 5, which is not referenced in the mention of sequentiality. If simultaneous operation consistently fails or has vastly lowered fidelity, crosstalk may be a problem for this architecture and it would be good to include more discussion of this important topic, as it may be make-or-break.

5) Two inner qubits (3 and 4) need to be initialised by QND readout via two others, which are then themselves read out via parity measurements. Does this method scale if there are 8 qubits in the array or will it need more sensors? In general, I would appreciate a more thorough discussion of scaling, whether in the supplementary or (preferably) in the main text. For example, how does the authors' design, the readout, initialization and operation method, scale to the second dimension? What are the limitations?

6) It is good to see that using virtual barriers, minimal residual exchange is achieved. However, the values of J34 and J56 are lower than the others, is this understood or simply due to a specific DC tuning?

7) What is the accuracy of the real-time feedback process? Does this contribute to SPAM errors?

8) For the GHZ state, the SPAM-removed fidelities are much higher (though not extremely high) than the uncorrected fidelities, so that it seems that SPAM errors make a large difference for the GHZ experiment, larger than for the two-qubit Bell state experiment. Is this related to the real-time feedback process, or because of the involvement of Q3 and Q4 in each operation?

Some comments related to references:

Quite a few titles in the references have errors in punctuation and capitalisation, this should be fixed.

The authors should cite other efforts to scale quantum dots, in one dimension (Weinstein et al., arXiv: 2202.03605) and two dimensions (Mortemousque et al., Nat. Nano, and Fedele et. Al, PRX Quantum).

...and wording:

Page 1, line 31, should be significantly “compromised” and not “comprised”.

Line 240, references Ref. [37], which is previous work by the authors; this states Bell fidelities between 78-89%, which is indeed less than the SPAM-corrected fidelities reported here, but I could not find at a glance in the previous paper whether those were SPAM-corrected. If they were not, then I would remove the wording “considerably higher” as the uncorrected fidelities in the current manuscript are not that much higher at all.

P. 21, figure caption, Extended “Date” should be “Data”

Finally, about the Methods and Supplementary:

In Methods, section “Automated Calibration Routines, it is stated “Fig 1d list” ...but Figure 1d does not seem to state what is references, unless I have misunderstood? Either way, it is unclear. Also, it should be Fig. 1d “lists”.

6) The comments on the other devices fabricated are very useful, as the field generally can suffer from the phenomenon of the “hero device”. Were these three devices the only ones fabricated, or only the ones also measured?

7) What limits T_2^* ? Is it demonstrated that it is the micromagnet coupling in charge noise? It may be quite a low number, but it looks remarkably consistent across 12 qubits in two devices. Is there a way forward to reduce this?

Extended Data Figure 4: Panel j is fit to a line?

Supplementary Fig 5: Why is there no mesa in the design, given that this then requires the addition of a silicon nitride layer and added effort?

Referee #3 (Remarks to the Author):

This manuscript reports on experimental progress in silicon-based spin qubit devices, with 6 fully controllable spin qubits with reasonably good fidelities. Semiconductor spin qubits have been lagging behind other qubit platforms, but they may be more suitable for larger-scale quantum computing devices and recent progress in this field has been very promising. This work uses known methods - multi-layer overlapping gates and micromagnet - to build the device and introduce novel protocols for initialization and readout with significant improvement on previously reported results.

The main appealing point of this work is that it demonstrates that it is possible to have all the ingredients of quantum computation in a single device with good fidelities while increasing the number of qubits. Previous work by others (e.g. Ref [21]) has the same number of or more quantum dots with single occupancy with electron spins, but full control of all the QD spin qubits has not been demonstrated.

The device in this work is similarly a linear array of quantum dots with a micromagnet for individual addressability. Full control over all 6 qubits is demonstrated by creating entangled Bell states and GHZ states with various combinations of qubits. They utilize a new initialization/readout scheme that uses the initialization by measurement with real-time feedback. It is sophisticated but straightforward, and it can be applied to other spin qubit devices.

This manuscript presents impressive results and it is a big step toward larger scalable spin qubit devices. The manuscript is well written with a lot of detailed information on the experiments. I think it is worth publication in Nature after the authors address some concerns and comments below.

1. I think scalability is still an issue. This 6-qubit device with full controllability is impressive, but the current methodology may not be suitable for even larger systems. I would like the authors to comment on some of the related issues below and provide some perspectives on how we can scale up the spin qubit devices in this architecture.

- Use of a micromagnet to create the magnetic field gradient for EDSR seems to be difficult to scale up.
- Single qubit gates were limited to sequential operations. Parallel operation of single-qubit gates will be essential for larger systems.
- Sensing dots are at both ends of the linear array. It limits the accessibility of the sensing dots to the individual dots in the middle region. It is in contrast to other similar linear quantum dot devices where the sensing dots are in parallel to the array. Is there any particular reason for choosing this architecture?

2. This paper reports on the fidelities of entangled states (Bell state and GHZ state), but does not cite fidelities for initialization/readout and two-qubit gates. Only single qubit RB fidelity is reported. It seems to be straightforward to run some benchmark protocols to characterize the fidelities of individual operations.

3. Valley degeneracy is a big issue for any electron qubits in silicon. The valley splitting in this device is somewhat larger than typical values. Any explanation?

Author Rebuttals to Initial Comments:

Referee #1 (Remarks to the Author):

Review of the manuscript “Universal control of a six qubit quantum processor in silicon” by Stephan G. J. Philips et al., submitted to Nature.

In this work, the authors demonstrate the first semiconductor 6-qubit chip. For the first time, they combine single- and two-qubit gates with high-fidelity initialization and read-out strategies. Their demonstration sets a milestone for semiconductor spin qubits. The authors discuss how to operate and tune their dense linear array of qubits and point out that 108 parameters have to be tuned. The authors take many efforts to provide all details about this new large-scale tuning efforts and the related time-budget (methods and extended data Fig. 4 and in supplementary materials). Uncertainties are treated very well and data is robust. The applied methods are not new and the fidelities are not the highest compared to previous publications which just focus on one functional block or fidelities measured on small demonstrators (with smaller qubit number), but this cannot be expected when combining all functionalities in one multi-qubit device for the first time. The resonant frequency of the qubits did not turned out be as planned (the authors provide a good analysis in the supplementary material). Nevertheless, the authors could compensate this and manage to make the qubits nearly “identical” in terms of Rabi frequency and exchange interaction.

The manuscript is very clearly written and exemplary for transparency on all methods. The presentation quality is very high. In fact, it is amazing how many details about experimental setup, the sample design, heating effects, cross-coupling, sample evolution are provided by the authors. This paper is a highlight on spin-qubit quantum computing and one of the best I ever read. It is definitely recommend publication in Nature.

We thank the referee for reviewing our manuscript, for the very positive comments and for recommending publication in Nature. We also thank the referee for the helpful additional comments and suggestions below, to which we respond point by point (and we apologize for the typos).

Some minor suggestions/questions:

96: “Figure 1d” has to be changed to “Figure 1c”.

We have corrected the typo.

151: “Figure 2c” has to be changed to “Figure 2b”.

We have corrected the typo.

Caption Fig. 2: “qubit frequency for each the six qubits” change to “qubit frequency for each OF the six qubits”

We have corrected the mistake.

Caption Fig. 2: I suggest a more suitable title: “Experiment overview” could be replaced by “Single qubit gate characterization”

We have followed the referee’s suggestion.

Colours are hard to distinguish in Fig. 1d.

We have increased the contrast of the colours to resolve this problem.

Fig. 1a is flipped compared to Extended Data Fig. 7

We have rotated Extended Data Fig. 7 in order to match the orientation of Figure 1.

215: “Twice a week, we run...”. This information is repeated in the Method section.

We have reduced the overlap by adjusting the Methods section.

1. Extended Data Fig. 7 shows a laser diode. Was it required for the experiment?

We place laser diodes by default on the PCB's of our experiments as a mean to reset the sample (background charges are freed from the charge traps by the light from the laser diode when it is activated). For the experiments presented in this paper we did not use it, as the accumulation voltages were consistent and stable across the device (over multiple cooldowns). We added this information to the figure caption.

2. Is it correct that the Co micromagnets are not covered in order to prevent oxidization?

The magnets are indeed not covered. We have worried about oxidation of the Cobalt in the past but so far have not found it necessary to cover the micromagnets. We have added this information to the methods section on sample fabrication.

3. Does the 3-electron filling of the outer QD alter the expectation for the resonant frequency due to e.g. different g-factor?

This is a good question, we never explicitly tested this in the lab. We would expect some change, as the local electric field on the quantum dot increases with the number of electrons, and we expect that the resonance frequency can be slightly Stark shifted. In fact, this could possibly impact the distribution of resonance frequencies in Fig. 2b, although the effect is likely small. As we have not tested this, we prefer not to comment on this further in the manuscript.

4. Conclusion: For such an exemplary paper, the authors might elaborate more on (their opinion) how to scale-up spin qubit chips in the future. What are the main upcoming challenges? What has to be developed next?

We understand the importance of thinking about scaling and the main upcoming challenges. We stated at the end of the conclusion paragraph:

"We estimate that the concepts used here for control, initialization and readout can be used without substantial modification in arrays that are twice as long, as well as in small two-dimensional arrays. Scaling further will require additional elements such as cross-bar addressing or on-chip quantum links [43]."

Admittedly, this statement is rather brief and there are many relevant considerations that one could make. In fact, this constitutes a full research topic in itself, which we have spent considerable time thinking about and published several proposal/design papers, on dense 2D arrays (Li, Ruoyu, et al. *Science advances* 4.7 (2018)), on sparse 2D arrays (Boter, Jelmer M., et al., *arXiv preprint arXiv:2110.00189* (2021)) and networks of local registers (Vandersypen, L. M. K., et al. , *npj Quantum Information* 3.1 (2017)). Also other groups have given this topic much consideration, see e.g. Crawford, O., et al., *arXiv preprint arXiv:2201.02877* (2022), Veldhorst et al., *Nat Commun* 8, 1766 (2017) and Taylor et al. *Nature Phys.* 1, 177-183 (2005).

We have slightly expanded the concluding two sentences, and included additional references. Additionally, we provide more considerations on scaling in a new section in the Supplementary Information.

Referee #2 (Remarks to the Author):

The manuscript by Philips et al. reports on universal control achieved over a linear array of six quantum dots, forming six Loss-DiVincenzo spin qubits. The experiments performed are extremely technically challenging, and it is a feat to reliably perform the voltage pulses and calibrations required to operate these complex quantum gates in a multi-dimensional gate space. The manuscript is very

well written, and I congratulate the authors on getting all aspect of this difficult experiment to work. In fact, the main strength of the paper is in combining state-of-the-art initialisation, readout, and operation fidelities with consistent fabrication, FPGA-based improvements such as the active reset, and keeping all of this consistent while scaling to six quantum dots. In all, I think the current demonstration is a large step forward for the field of spin qubits and solid-state qubits. While it also posits a lot of questions (regarding maintenance of coherence, gate fidelity as the number of qubits is scaled, and design and routing scaling into the second and third dimension), these are topics crucial to explore and therefore the current work will open more areas of further research in the field, and should be brought to the notice of the wider community. Therefore I would recommend publication in Nature.

We thank the referee for reviewing our manuscript, for the very positive comments and for recommending publication in Nature. We also thank the referee for the helpful additional comments and suggestions below, to which we respond point by point (and we apologize for the typos).

I have some comments and questions I would like the authors to address:

1) The authors mention that they have solved the previous problem of low valley splittings, but I could not find an explanation how. Could they comment on this? One of the devices measured still suffered from this issue, so it seems to be still relevant.

We observe on average larger valley splittings compared to 5 years ago as a result of the work of the Scappucci group and others, see e.g. Wuetz et al, <https://arxiv.org/pdf/2112.09606.pdf>.

As the values of the valley splittings are still dependent on a statistical distribution, it is still possible to have lower values on one or more dots. For instance, in device A, we appear to have been unlucky that e.g. an atomic step at the interface at the location of qubit 3 may have caused a small valley energy. We would therefore not claim to have fully solved this problem, but certainly there is good progress, and ideas in the literature suggest that further progress is possible, see again e.g.

<https://arxiv.org/pdf/2112.09606.pdf> as well as

<https://journals.aps.org/prb/abstract/10.1103/PhysRevB.104.085406>.

We added a reference to the used substrates to supplementary table 1.

2) The caption of Fig. 1 contains the text "...up to spin relaxation"; what is this value of overall spin relaxation time, especially for the outer spins? I could not find a reference to this, only summer lower bounds. Does it vary across the array? Also, in Fig. 1e, the spin symbols and their colour are unclear at first glance and could be explained in the caption. Also, this section should refer to Extended Data Figure 2 (this is currently referred to only in the text) as the caption should be standalone, and it is hard to parse without referring to EDF2.

Indeed we have only measured a loose lower bound on T_1 : on a millisecond timescale, we observe no indication of spin relaxation. Given that the wait time between the end of the readout segment and the start of the next initialization segment is only 10's of microseconds, it is safe to neglect spin relaxation. Triggered by the referee's questions, we have therefore dropped the text "up to spin relaxation". Pragmatically speaking, we know that the T_1 's are long enough that we don't need to worry about spin relaxation.

To avoid confusion, we have changed all spin symbols to black color, as the colors had no particular meaning.

We thank the referee for the suggestion to add a reference to Extended Data Figure 2, and have added it to the caption.

3) The parity readout method is clever in solving the problem of the sensors being insensitive to the inner dots, but post-selection means many shots could be disregarded. What are the implications of

this for the operation of the processor and how much overhead does this add to the operation? How many runs are typically discarded?

Just to be sure, the QND readout method solves the problem of the sensors being insensitive to the inner dots, whereas post-selection on the parity readout indeed means a subset of the shots is discarded. In addition, the QND readout (e.g. of qubit 3) here also relies on parity readout (of qubits 1 and 2). We comment on both aspects below:

- Implications of using QND readout: electrons on the inside of an array (too far from the sensors to be read out directly) can be read out by mapping their state to that of electrons closer to the sensors. Alternative methods would typically involve shuttling electrons through the array, which is not necessary when using the present QND readout method.
- Overhead associated with discarding shots : When using parity readout on a single pair, 5% of the runs are discarded on average (95% success rate, as stated in the main text). In case two pairs are used, about 10% are discarded ($1-0.95^2$) as part of the initialization procedure. When using repeated QND measurements for 6 qubits, many more initializations are needed and we discard ~65% of the data set. This is more or less what is expected, as typically 16-24 parity measurements (including the QND repetitions and the measurements used for state preparation) are performed when working with all six qubits. For 20 measurements each of which keeps 95% of the shots, $1-0.95^{20} = 64\%$ of the shots will be kept on average. When we do not discard any runs, the initialization fidelity is reduced by ~6-9%. Finally, we point out that during the readout stage, the fidelity of the parity readout of the first two and the last two qubits is not affected if we do not discard any runs, but the QND readout fidelities are affected. We have added this information to the methods section.

4) The paper generally performs readout and operation sequentially. The authors should clearly explain why this is not performed simultaneously? (I assume frequency multiplexing is possible as this is used for the EDSR antenna.) Is this limited by readout crosstalk? For operation, the fact that crosstalk is an issue is similarly buried in Extended Data Figure 5, which is not referenced in the mention of sequentiality. If simultaneous operation consistently fails or has vastly lowered fidelity, crosstalk may be a problem for this architecture and it would be good to include more discussion of this important topic, as it may be make-or-break.

We structure our response by discussing separately the case of single-qubit gates, two-qubit gates and readout:

- Single qubit gates: there is no fundamental issue stopping us from performing multiple single-qubit gates at the same time. We chose not to do this in this first six-qubit experiment, as it would significantly increase the number of variables that needs to be managed in order to account for the crosstalk effects we have observed. We have added the following after "... to keep the calibration simple" in the main text: "Simultaneous rotations would involve additional characterization and compensation of cross-talk effects, see also Extended Data Fig. 5."
- Two qubit gates: Since sequential operation is easier at this stage, we performed the two-qubit gates sequentially. However, we do not expect a fundamental issue with simultaneous two-qubit gates either, as long as there is at least one idling dot in between. Also here, additional calibrations would be required, given that capacitive cross-talk effects will not be perfectly taken care of by the use of virtual gates. Simultaneous two-qubit gates without an idling dot in between would need to be slower. The reason is that we pulse not only the barrier between the target electrons of the two-qubit gate, but also the barriers on the outside of the target pair in order to reach a sufficiently strong exchange interaction. In the concluding paragraph, we added a subsentence as follows "The use of simultaneous single-qubit rotations and simultaneous two-qubit CZ gates will keep pulse sequences more compact, at the expense of additional calibrations."
- Readout: there is no direct crosstalk between the readout signals of both resonators (the frequencies of the two tank circuits are sufficiently separated). Additionally, the signal from the

left electron pair in the right sensing dot is negligible and vice versa. We therefore expect simultaneous readout would work well.

The main reason we did not read out the signal from both resonators in parallel is that the readout time was not limiting (other problems required more attention). Furthermore, the code presently used for the measurements doesn't have good support for performing parallel readout (this could of course be implemented). We have added this information to the caption of Ext. Data Fig. 2.

5) Two inner qubits (3 and 4) need to be initialised by QND readout via two others, which are then themselves read out via parity measurements. Does this method scale if there are 8 qubits in the array or will it need more sensors? In general, I would appreciate a more thorough discussion of scaling, whether in the supplementary or (preferably) in the main text. For example, how does the authors' design, the readout, initialization and operation method, scale to the second dimension? What are the limitations?

In principle one can perform QND readout of qubit 4 by mapping its state onto that of qubit 3, and from there onto that of qubit 2, which is in turn read out using the charge sensor. QND readout of qubit 5 can be done by mapping it onto the state of qubit 4, and so on. This would involve linear overhead in the length of the chain. However, the errors from the successive mapping (CROT) operations would accumulate for concatenated QND readout. Alternatively, the number of QND repetitions needed to achieve the target readout fidelity would increase, making the overall overhead a bit higher than linear in the distance. This concept works also in 2D, again with overhead.

Regarding the broader question, we stated at the end of the concluding paragraph

“We estimate that the concepts used here for control, initialization and readout can be used without substantial modification in arrays that are twice as long, as well as in small two-dimensional arrays. Scaling further will require additional elements such as cross-bar addressing or on-chip quantum links [43].”

Concretely, we would have in mind to add charge sensors alongside extended linear arrays, as in device A and in Zajac et al., *Phys. Rev. Applied* 6, 054013 (2016). The micromagnet design would be modified for small 2D arrays in order to get selectively of the qubit frequencies in two dimensions. While we understand the importance of thinking about scaling to two dimensions and what the limitations are, there are so many relevant considerations that this constitutes a full research topic in itself. In fact, we have spent considerable time thinking about this topic and published several proposal/design papers on dense 2D arrays (Li, Ruoyu, et al. *Science advances* 4.7 (2018)), on sparse 2D arrays (Boter, Jelmer M., et al., *arXiv preprint arXiv:2110.00189* (2021).) and networks of local registers (Vandersypen, L. M. K., et al. , *npj Quantum Information* 3.1 (2017)). Also other groups have given this topic much consideration, see e.g. Crawford, O., et al., *arXiv preprint arXiv:2201.02877* (2022) and Veldhorst et al., *Nat Commun* 8, 1766 (2017)

We have slightly expanded the concluding two sentences, and included additional references, and additionally we provide more considerations in a new section in the Supplementary Information.

6) It is good to see that using virtual barriers, minimal residual exchange is achieved. However, the values of J34 and J56 are lower than the others, is this understood or simply due to a specific DC tuning?

Based on the numbers in the table of Fig. 3g, we presume the referee asks why the “on” values of J34 and J56 are lower than the others. The reasons are as follows:

- J56 : gate B6 was not connected to a fast gate, which meant we could not enhance J56 as much by pulsing, as the exchange coupling strength for the other pairs. This can also be seen in Fig. 2h, where the slope for J56 is the lowest of all exchange couplings.
- J34: this was indeed due to the DC tuning, by choice. We did not want to push to a high value of J as the Zeeman energy difference between qubits 3 and 4 was rather small (~30 MHz) and for the CPhase gate, we need the Zeeman energy difference to exceed J by a factor of ~10.

7) What is the accuracy of the real-time feedback process? Does this contribute to SPAM errors?

The accuracy is about 95%, see the statement in the main text:

“If the qubit initialization is successful, the second measurement should return an odd parity (typically 95% success rate).”). The limitation is mainly in the ability to adiabatically separate the electrons after PSB readout.

It should not contribute to SPAM errors as we post-selected the data. Without postselection, it would contribute to SPAM errors, see also the response to point 3 of this referee and the new paragraph in the Methods section.

8) For the GHZ state, the SPAM-removed fidelities are much higher (though not extremely high) than the uncorrected fidelities, so that it seems that SPAM errors make a large difference for the GHZ experiment, larger than for the two-qubit Bell state experiment. Is this related to the real-time feedback process, or because of the involvement of Q3 and Q4 in each operation?

This is related to an increase in readout errors when operating multiple qubit pairs at the same time.

This effect is demonstrated in the Extended Data Fig. 3 (and also comes back in the response to point 3 of this referee). We clarified this point in the main text.

Some comments related to references:

Quite a few titles in the references have errors in punctuation and capitalisation, this should be fixed.

We have corrected the errors in the references.

The authors should cite other efforts to scale quantum dots, in one dimension (Weinstein et al., arXiv: 2202.03605) and two dimensions (Mortemousque et al., Nat. Nano, and Fedele et. Al, PRX Quantum).

We agree that these are relevant references and added them to the introduction.

...and wording:

Page 1, line 31, should be significantly “compromised” and not “comprised”.

We have corrected the typo.

Line 240, references Ref. [38], which is previous work by the authors; this states Bell fidelities between 78-89%, which is indeed less than the SPAM-corrected fidelities reported here, but I could not find at a glance in the previous paper whether those were SPAM-corrected. If they were not, then I would remove the wording “considerably higher” as the uncorrected fidelities in the current manuscript are not that much higher at all.

These fidelities were SPAM corrected. SPAM errors in Ref. [38] are [92/54] for Q1 and [86/76] for Q2, where the first and second number refer to the readout fidelity for spin-down and spin-up respectively. The difference in the uncorrected Bell state fidelities between [38] and the present work is thus even larger than the difference in the corrected Bell state fidelities.

We clarified in the manuscript that the fidelities from [38] are corrected for SPAM errors.

P. 21, figure caption, Extended “Date” should be “Data”

We have corrected the typo.

Finally, about the Methods and Supplementary:

In Methods, section “Automated Calibration Routines”, it is stated “Fig 1d list”...but Figure 1d does not seem to state what is referenced, unless I have misunderstood? Either way, it is unclear. Also, it should be Fig. 1d “lists”.

We have corrected the typo and reference to figure 1f.

6) The comments on the other devices fabricated are very useful, as the field generally can suffer from the phenomenon of the “hero device”. Were these three devices the only ones fabricated, or only the ones also measured?

Many more devices have been screened using dipstick (4K) measurements. We screen devices by checking for leakage, accumulation below every gate and stability of the device. Usually 12 to 24 devices are fabricated in a single batch, which are all screened. The best device (if meeting our requirements) is selected and cooled down in a dilution refrigerator. The yield of our batches ranges in between 0-50% (we also had several batches with no working devices). We added this information to the Methods section on device fabrication.

7) What limits T_2^* ? Is it demonstrated that it is the micromagnet coupling in charge noise? It may be quite a low number, but it looks remarkably consistent across 12 qubits in two devices. Is there a way forward to reduce this?

We suspect the qubits are limited by charge noise that couples in by the micromagnet, as the T_2^* is below the expected value for isotopically enriched ^{28}Si . We did not perform any experiment targeted at confirming this.

If the short T_2^* times are caused by the micromagnets, the coherence time could be extended by:

- Reducing the decohering longitudinal gradient fields of the micromagnet (see supplementary figure 1)
- Optimising the spin qubit stack, such that there is less charge noise (e.g. passivation of defects at the interfaces).

Since we did not focus on investigating the decoherence mechanism, it is our preference not to speculate on it.

Extended Data Figure 4: Panel j is fit to a line?

The data in panel j is fitted to a line, plotting the data on a semilog plot. This increases the robustness of the fit compared to fitting to an exponential on a linear plot. We replotted this panel on a log-scale to clarify this.

Supplementary Fig 5: Why is there no mesa in the design, given that this then requires the addition of a silicon nitride layer and added effort?

We could have used a mesa instead and the choice is partly based on historical reasons. We note however that fabricating a mesa also involves additional e-beam lithography steps. First the mesa itself needs to be patterned. Second, whereas the fine gates must be made from thin metal, stepping over the sharp etched edge of the mesa requires thick gate metal, which must be separately patterned. In contrast, climbing onto the silicon nitride layer is feasible with thin metal, since the edge is smooth.

Referee #3 (Remarks to the Author):

This manuscript reports on experimental progress in silicon-based spin qubit devices, with 6 fully controllable spin qubits with reasonably good fidelities. Semiconductor spin qubits have been lagging behind other qubit platforms, but they may be more suitable for larger-scale quantum computing

devices and recent progress in this field has been very promising. This work uses known methods - multi-layer overlapping gates and micromagnet - to build the device and introduce novel protocols for initialization and readout with significant improvement on previously reported results.

The main appealing point of this work is that it demonstrates that it is possible to have all the ingredients of quantum computation in a single device with good fidelities while increasing the number of qubits. Previous work by others (e.g. Ref [21]) has the same number of or more quantum dots with single occupancy with electron spins, but full control of all the QD spin qubits has not been demonstrated.

The device in this work is similarly a linear array of quantum dots with a micromagnet for individual addressability. Full control over all 6 qubits is demonstrated by creating entangled Bell states and GHZ states with various combinations of qubits. They utilize a new initialization/readout scheme that uses the initialization by measurement with real-time feedback. It is sophisticated but straightforward, and it can be applied to other spin qubit devices.

This manuscript presents impressive results and it is a big step toward larger scalable spin qubit devices. The manuscript is well written with a lot of detailed information on the experiments. I think it is worth publication in Nature after the authors address some concerns and comments below.

We thank the referee for reviewing our manuscript, for the very positive comments and for recommending publication in Nature. We also thank the referee for the helpful additional comments and suggestions below, to which we respond point by point.

1. I think scalability is still an issue. This 6-qubit device with full controllability is impressive, but the current methodology may not be suitable for even larger systems. I would like the authors to comment on some of the related issues below and provide some perspectives on how we can scale up the spin qubit devices in this architecture.

- Use of a micromagnet to create the magnetic field gradient for EDSR seems to be difficult to scale up.
- Single qubit gates were limited to sequential operations. Parallel operation of single-qubit gates will be essential for larger systems.
- Sensing dots are at both ends of the linear array. It limits the accessibility of the sensing dots to the individual dots in the middle region. It is in contrast to other similar linear quantum dot devices where the sensing dots are in parallel to the array. Is there any particular reason for choosing this architecture?

Indeed, multiple issues remain and the current methodology is not sufficient to enable very large arrays. Therefore, we stated at the end of the concluding paragraph

“We estimate that the concepts used here for control, initialization and readout can be used without substantial modification in arrays that are twice as long, as well as in small two-dimensional arrays. Scaling further will require additional elements such as cross-bar addressing or on-chip quantum links [43].”

Regarding EDSR, the issues are that 1) the artificial spin orbit coupling generated by the micromagnets allows charge noise to couple to the spin qubits, 2) no architecture has been shown yet that supports a design for micromagnets that easily scales in two dimensions and 3) questions remain on how uniform or reproducible the field of a micromagnet can be (across a chip with thousands of micromagnets). Therefore, we believe that a more fruitful path is to use local arrays of a moderate size (where micromagnets can be used and designed to minimize decoherence gradients) and network them together on the same chip using quantum links. We have slightly expanded the

concluding two sentences, and included additional references, and additionally provide more considerations in a novel section in the Supplementary Information.

With respect to parallel operation of single-qubit gates, we had written in the concluding paragraph:

“The use of simultaneous single-qubit rotations and simultaneous two-qubit CZ gates will keep pulse sequences more compact.”

For single-qubit gates, there is no fundamental issue stopping us from performing multiple single qubit gates at the same time. We chose not to do this in this first six-qubit experiment, as it would significantly increase the number of variables that needs to be managed in order to account for the crosstalk effects, especially in light of crosstalk effects. We have added the following after “... to keep the calibration simple“ in the main text: “(simultaneous rotations would involve additional characterization and compensation of cross-talk effects, see also Extended Data Fig. 5)”.

Finally, regarding the sensing dots, device A in fact had a sensing dot near the middle of the array in addition to two sensing dots at the end of the array. The sensing dots at the end of the array have the maximum sensitivity to interdot transitions, which we are interested in here. Since we found the full 6-qubit array could be tuned up without the third sensing dot alongside the array, we left it out in devices B and C in order to slightly reduce the complexity of fabrication. There is no real obstacle to putting it back. Such a third sensor would have made it possible to probe PSB directly on the inner pair of dots. This being said, mapping the state of qubits 3 and 4 onto the outer pairs has worked very well and this concept can prove very useful in larger arrays, especially 2D arrays. This is also captured in the new section on scaling in the Supplementary Information.

2. This paper reports on the fidelities of entangled states (Bell state and GHZ state), but does not cite fidelities for initialization/readout and two-qubit gates. Only single qubit RB fidelity is reported. It seems to be straightforward to run some benchmark protocols to characterise the fidelities of individual operations.

It is straightforward on paper to run two-qubit randomized benchmarking protocols, although in previous experiments we have found it can be subtle in practice (see e.g. X. Xue et al, Phys. Rev. X **9**, 021011, 2019). While we didn't attempt this, we think two-qubit randomized benchmarking on the present sample would have been possible. In practice one can only perform so many experiments and we feel that we have already performed a very comprehensive study of the six-qubit device, from initialization to readout and control.

Instead of initialization and readout fidelities, we provide visibilities which capture both together. Unfortunately, it is not trivial to separate the readout errors from the initialization errors without making additional assumptions on the error mechanisms. To add to the complexity, we initialize by measurement and the repeated measurements additionally involve initialization steps, so readout and measurement are quite convoluted in this experiment.

3. Valley degeneracy is a big issue for any electron qubits in silicon. The valley splitting in this device is somewhat larger than typical values. Any explanation?

The valley splitting is indeed somewhat larger than in most prior reports. We observe on average larger valley splittings compared to 5 years ago as a result of the work of the Scappucci group and others, see e.g. Wuetz et al, <https://arxiv.org/pdf/2112.09606.pdf>. However, in the same preprint, a distribution for the expected valley splittings can be found, and the values in device C are mostly on the high side of that distribution. For comparison, supplementary table 1 shows lower valley splittings

for devices A and B. The explanation of the larger than typical valley splittings may be in part improved heterostructures and in part statistical variations and some luck for device C.